# Critical scaling of whole-brain resting-state dynamics

Adrián Ponce-Alvarez [1,2✉], Morten L. Kringelbach [3,4,5] & Gustavo Deco [1,6]

Scale invariance is a characteristic of neural activity. How this property emerges from neural interactions remains a fundamental question. Here, we studied the relation between scale-invariant brain dynamics and structural connectivity by analyzing human resting-state (rs-) fMRI signals, together with diffusion MRI (dMRI) connectivity and its approximation as an exponentially decaying function of the distance between brain regions. We analyzed the rs-fMRI dynamics using functional connectivity and a recently proposed phenomenological renormalization group (PRG) method that tracks the change of collective activity after successive coarse-graining at different scales. We found that brain dynamics display power-law correlations and power-law scaling as a function of PRG coarse-graining based on functional or structural connectivity. Moreover, we modeled the brain activity using a network of spins interacting through large-scale connectivity and presenting a phase transition between ordered and disordered phases. Within this simple model, we found that the observed scaling features were likely to emerge from critical dynamics and connections exponentially decaying with distance. In conclusion, our study tests the PRG method using large-scale brain activity and theoretical models and suggests that scaling of rs-fMRI activity relates to criticality.

[1] Center for Brain and Cognition, Computational Neuroscience Group, Department of Information and Communication Technologies, Universitat Pompeu Fabra, Barcelona 08005, Spain. [2] Departament de Matemàtiques, Universitat Politècnica de Catalunya, Barcelona, Spain. [3] Department of Psychiatry, University of Oxford, Oxford OX3 7JX, UK. [4] Center for Music in the Brain, Department of Clinical Medicine, Aarhus University, Aarhus 8000, Denmark. [5] Centre for Eudaimonia and Human Flourishing, Linacre College, University of Oxford, Oxford, UK. [6] Institució Catalana de la Recerca i Estudis Avançats (ICREA), Barcelona 08010, Spain. ✉email: adrian.ponce@upc.edu

nteresting phenomena in biological systems are often collective behaviors emerging from the interactions among their constituents. Growing evidence indicates that large-scale spontaneous brain activity is an emergent phenomenon continuously generating patterned activity at multiple spatiotemporal scales[1–3]. How patterned activity arise from the brain's connectivity is a fundamental question in neuroscience. A hallmark of spontaneous, or resting-state (rs), whole-brain activity is scale invariance. Indeed, scale-free (power-law) power spectrum has been reported in human brain dynamics acquired using fMRI[4,5], electrocorticography[6], LFPs[7], and MEG[8]. Moreover, scale-invariance of propagating clusters of activity (*neuronal avalanches*) has been observed in human fMRI recordings[9], M/EEG fluctuations[10,11], and zebrafish whole-brain calcium imaging[12]. These works add up to a large body of studies showing scale-invariant neuronal avalanches at the microcircuit level[13–18].

In physical systems, scale-invariance is observed at critical points. Thus, the observation of power-law statistics in neural activity has contributed to support the idea that spontaneous neural activity operates close to a phase transition[19]. Several studies have shown that critical neural systems maximize information transmission, storage, and processing[12,20–22]. Interestingly, it has been shown that brain activity deviates from critical-like behavior in different brain states and neuropsychiatric disorders[23–29]. For instance, using fMRI[30,31] and voltage imaging[32], it has been shown that awake resting state displays critical-like dynamics, yielding maximal information capacity and transmission, while anesthesia states depart from criticality. For these reasons it is believed that critical dynamics are a characteristic of healthy, awake spontaneous neural activity.

Because the critical behavior of a physical system is governed by fluctuations that are statistically self-similar, its statistics are re-scaled after gradual elimination of the correlated degrees of freedom. This is achieved through the Renormalization Group (RG) procedure. This method tracks the change of the joint probability distribution of the system variables after successive coarse-graining at different scales. In the case of critical systems at equilibrium, probability distributions are scale-invariant under iterated coarse-graining and represent fixed points of the RG. In the case of neural activity, to account for the (unknown) topology of interactions, a phenomenological renormalization group (PRG) procedure has been proposed in which maximally correlated variables are grouped together[33], instead of locally grouping variables that are spatially close as in most applications of RG. This method successfully revealed scaling features in local single-neuron recordings from the mouse hippocampus[33,34] and, recently, in other areas of the mouse brain[35]. The method has been previously tested on theoretical models[36], but it remains to be tested at the large scale and combining theoretical models and data.

Moreover, how the scaling properties of brain dynamics relate to structural principles remains unclear. Previous investigations using retrograde tract tracing methods in mice and nonhuman primates' cortices[37–39] have shown that the probability of a connection existing between two given cortical areas declines with distance. An exponential decay with interareal distance, known as exponential distance rule (EDR), has been proposed as a simple, geometrically-constraint wiring principle[38–41]. Nevertheless, a recent study using fluorescent mapping of neuronal projections found that a power law decay was a marginally better fit than exponential decay[42]. Special attention has been paid to long-range connections that deviate from simple connection decay as a function of distance, in terms of their impact on wiring-cost[43] and on dynamics of oscillator models[44,45].

Here, we studied how scale-invariance of brain dynamics relates to structural connectivity. For this, we examined the relation between human rs-fMRI signals and diffusion MRI (dMRI) structural connectivity, focusing on spatial anatomical constrains. Next, we studied the scaling properties of rs-fMRI dynamics using the correlation function and the PRG method. Finally, using a simple spin model with a geometrically-constraint connectivity, we showed that scaling in rs-fMRI signals is suggestive of critical behavior.

## Results

**Linear prediction of fMRI signals from structural connections.** We analyzed fMRI signals in a parcellation of $N = 1000$ regions of interest (ROIs) or nodes. The dataset was composed of 1003 individual scans of $n_F = 1200$ time frames. Structural connectivity was obtained using dMRI and probabilistic tractography in the same parcellation, resulting in a $N \times N$ coupling matrix $\boldsymbol{C}$ (see "Methods"). The data was obtained from the Human Connectome Project (HCP) public database. In this section, we studied the relation between fMRI dynamics and structural couplings.

We first note that the relation between the connectivity weight between two nodes and the Euclidean spatial distance between those nodes was approximately a power law (Fig. 1a), indicating the presence of long-range connections. To evaluate the importance of long-range connections, we compared the dMRI connectivity to a model connectivity based on EDR, i.e., $C_{ij} \propto \exp(-\gamma r_{ij})$, where $r_{ij} = \left| \vec{x}_i - \vec{x}_j \right|$, and $\vec{x}_i$ and $\vec{x}_j$ are the positions in 3D space of the centers of ROIs $i$ and $j$, respectively. Using least squares and for distances <50 mm, we obtained: $\gamma = 0.106 \pm 0.007$ mm$^{-1}$. Notably, the value of $\gamma$ is consistent with the extrapolation to human brain based on the relation between the EDR and the brain volume[39]. We noted that the EDR fit was higher for intra-hemispheric connections than for the long-range inter-hemispheric ones (Fig. 1b): the correlation between EDR and dMRI connectivities was equal to 0.69 and 0.49 for intra- and inter-hemispheric connections, respectively. By construction, dMRI and EDR connectivity matrices are symmetric.

We next tested the linear signal prediction of both connectivity matrices. Let $\boldsymbol{X}$ be the $N \times n_F$ data matrix containing the fMRI signals. Assuming linear couplings, we calculated the predicted signals as $\boldsymbol{X}_{\text{pred}} = \boldsymbol{CX}$ (i.e., the prediction of each signal given the rest of the network and couplings $\boldsymbol{C}$). The goodness of the linear prediction was given by the Pearson correlation between $X_{\text{pred}}(t)$ and $X(t)$ for all nodes and all subjects. Correlation coefficients were remarkably similar for the dMRI and the EDR, with means equal to $0.51 \pm 0.01$ and $0.49 \pm 0.01$, respectively (Fig. 1c, d). These average values were significantly higher ($p < 0.001$, Welch's $t$-test) than the one obtained using a shuffled connectivity that preserves the distribution of dMRI weights but destroys their spatial organization (mean correlation: $0.32 \pm 0.01$). Consistently, we noted that nodes for which the linear prediction was the lowest were those nodes that were weakly connected to the network, i.e., nodes with low node strength, where the strength of node $i$ is the sum of the weights of its connections with other nodes in the network, i.e., $\sum_{j \neq i} C_{ij}$ (Fig. 1e). Finally, we also tested (i) the linear prediction of each signal given the rest of signals from the same hemisphere and intra-hemispheric couplings and (ii) the linear prediction of each signal given the signals from the contralateral hemisphere and inter-hemispheric couplings (Supplementary Fig. S1). We found that, although inter-hemispheric predictions were reduced with respect to intra-hemispheric ones, they remained significant and were practically indistinguishable using the dMRI and the EDR connectivity matrices. Altogether, we concluded that both the dMRI and the

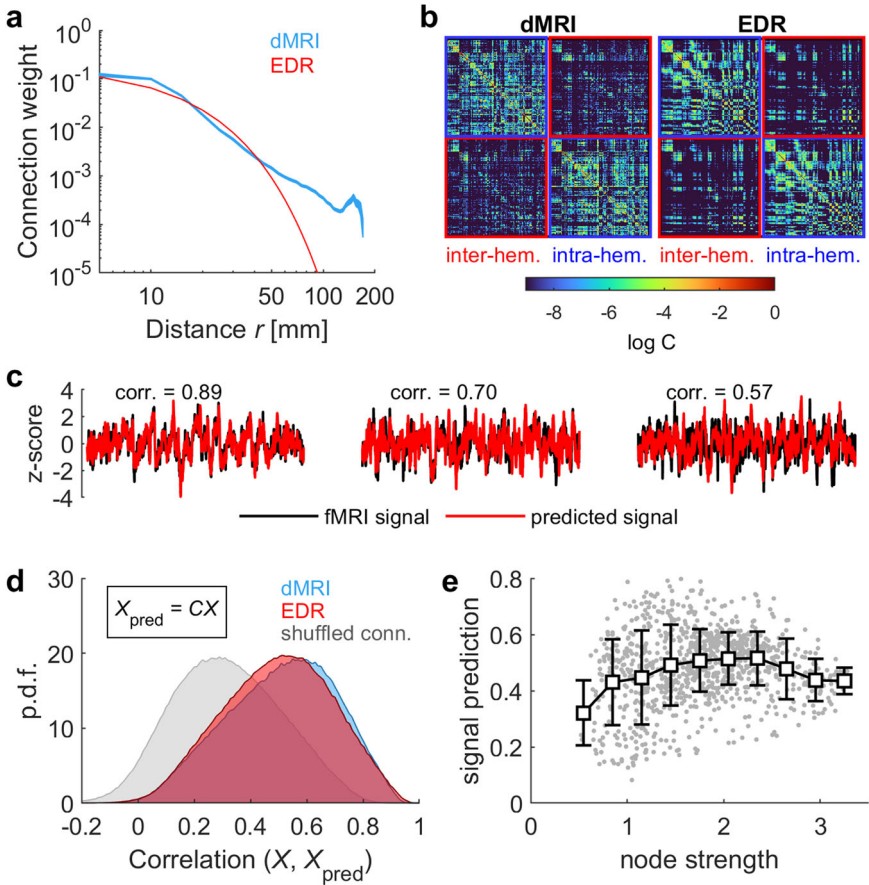

**Fig. 1 Linear prediction of fMRI signals using dMRI and EDR coupling matrices. a** dMRI weights as a function of the distance between pairs of nodes (blue) and the exponential approximation (red). The thickness of the blue trace indicates SEM. **b** Large-scale connectivity matrices: dMRI (left), EDR (right). Connectivity weights are presented in logarithmic scale. Portions of the matrices corresponding to intra- and inter-hemispheric connections are highlighted in *red* and *blue*, respectively. **c** Three example fMRI signals and their corresponding linear predictions using the EDR connectivity, i.e., $X_{\text{pred}} = CX$. The correlation between the actual and the predicted signals is also indicated. **d** Distributions of signal predictions using the dMRI connectivity (*blue*), the EDR (*red*), and a shuffled connectivity that preserves the distribution of dMRI weights but destroys the spatial organization (*gray*) ($n = 1000 \times 1003$). **e** Average signal prediction as a function of the strength of the nodes of the EDR connectivity. Each dot represents a node ($n = 1000$); the blue trace represents the average relation. Error bars indicate SD.

EDR connectivity matrices were equally good linear predictors of the fMRI signals.

**Correlation function and phenomenological renormalization-group.** We next evaluated the relation $g(r)$ between functional correlations of pairs of ROIs and their distance. As required for the modelling in the next section, we used binarized fMRI signals (see "Methods"). Briefly, for each scan, the z-scored time-series of each ROI, $z_i(t)$ ($1 \leq i \leq N$), was binarized by imposing a threshold $\theta = 1$. Thus, at each time frame $t$, the collective activity was described by a binary vector $\vec{\sigma} = [\sigma_1, \ldots, \sigma_N]$, with $\sigma_i = 1$ if $z_i(t) > \theta$ and $\sigma_i = 0$ otherwise. Binarization of time-series has proven to effectively capture and compress fMRI large-scale collective dynamics[9,31,46].

Consistent with previous work[47], we found that the average functional correlation, across all ROI pairs and all subjects, was approximately power-law, i.e., $g(r) \sim r^{-\tilde{\eta}}$, with a power exponent equal to $\tilde{\eta} = 0.513 \pm 0.009$ (Fig. 2a–c). For each individual scan, we tested the power-law hypothesis against an exponential alternative by calculating the ratio between explained variances ($R_{EV}$) of least-squares fits using the two competing regression models. We found a ratio of ~1.2 of explained variances systematically in favor of the power-law hypothesis (Fig. 2d).

The exact value of the exponent $\tilde{\eta}$ depends on the cutoffs $[r_{\min}, r_{\max}]$ used to constrain the power-law fitting, but for a large region in the $[r_{\min}, r_{\max}]$ plane we found a good fitting of the power law (explained variance $R^2 > 0.95$) and an exponent around 0.52 (Fig. 2e, f).

Power-law correlations are a hallmark of critical systems, but neither a necessary nor a sufficient condition. Recently, a PRG approach has been proposed to identify scale-invariant activity in neural systems[33]. Within this method, the collective activity is iteratively coarse-grained by grouping maximally correlated variables (see "Methods"). At each coarse-graining step $k = 0, 1, \ldots, k_{\max}$, clusters of size $K = 2^k$ are built, resulting in a system of $N/K$ coarse-grained variables and successively ignoring degrees of freedom. We calculated several observables of the coarse-grained variables and studied their evolution as a function of $K$.

We found that the variance $V$ of coarse-grained variables scaled as a power of the cluster size, i.e., $V \sim K^{\tilde{\alpha}}$, with an average exponent $\langle \tilde{\alpha} \rangle = 1.574 \pm 0.002$ across subjects (Fig. 3a). This exponent lies in the region between linear ($\tilde{\alpha} = 1$) and quadratic growth ($\tilde{\alpha} = 2$), corresponding to uncorrelated and fully-correlated systems, respectively. The distribution of the value of the exponents $\tilde{\alpha}$ from individual scans is shown in Fig. 3e.

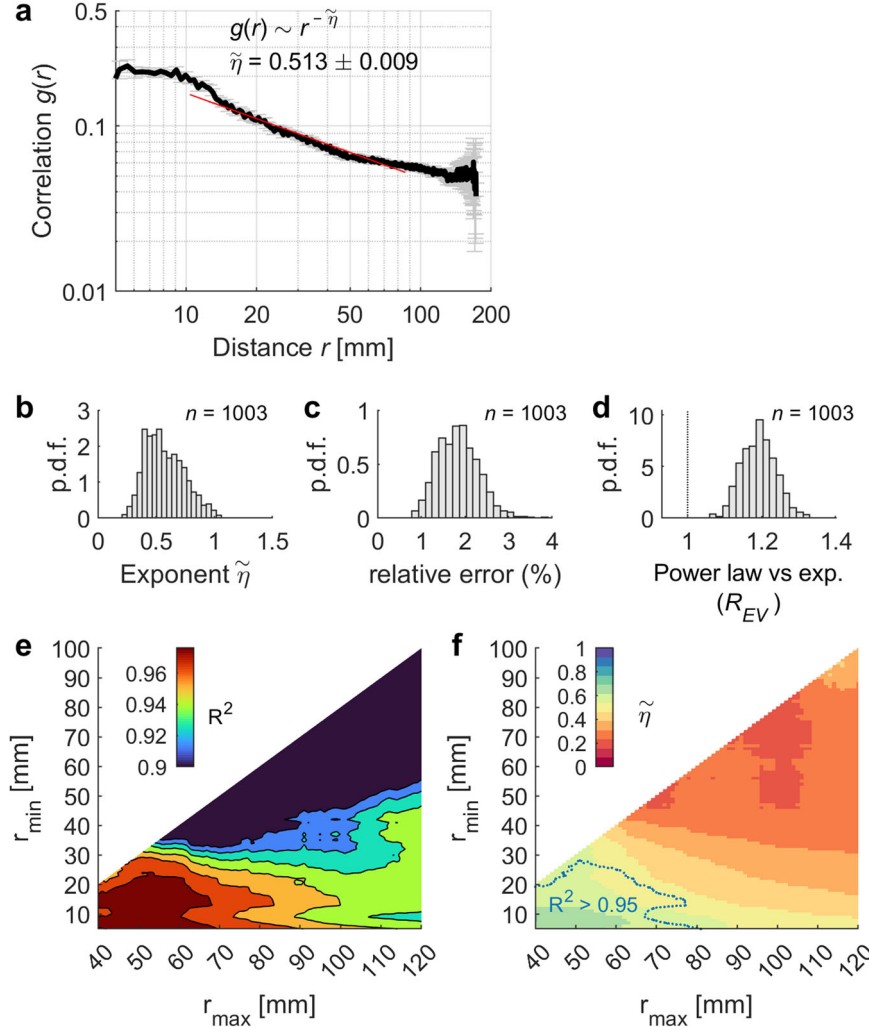

**Fig. 2 Correlation function. a** Correlation function of fMRI signals as a function of the distance between nodes. Error bars indicate SEM. The correlation function of fMRI signals was approximately power-law, i.e., $g(r) \sim r^{-\tilde{\eta}}$. The power law was fitted in the distance interval $r \in [10, 90]$ mm. **b** Distribution of the estimated power exponent for single-subject scans ($n = 1003$). **c** Distribution of the relative estimation error of exponent $\tilde{\eta}$, i.e., $\Delta\tilde{\eta}/\tilde{\eta}$, where $\Delta\tilde{\eta}$ is the least square estimation error of exponent $\tilde{\eta}$. Note that the average relative estimation error is <3%. **d** The power law fit was compared the one obtained using an exponential function by calculating the ratio between the explained variance of the competing regression models. Ratios larger than 1 favor the power law hypothesis. **e, f** When fitting the power law to $g(r)$ in the distance interval $r \in [r_{min}, r_{max}]$, for several combinations of $r_{min}$ and $r_{max}$, we found a large region in the $[r_{min}, r_{max}]$ plane with high explained variance $R^2$ (**e**) yielding power exponents $\tilde{\eta} \sim 0.52$ (**f**, the blue dotted line indicates the region for which $R^2 > 0.95$).

Individual exponents $\tilde{\alpha}$ had low uncertainty: on average, the exponent estimation error was equal to 0.03 (Fig. 3f), corresponding to an average relative estimation error equal to 1.99% (Fig. 3g). Moreover, the ratio between explained variances of least-square fits of $V(K)$ using a power law vs. an exponential distribution systematically favored the power-law hypothesis ($\langle R_{EV} \rangle = 1.34$; Fig. 3h).

Another interesting observable is the probability of silence activity $P_{silence}$, i.e., time frames in which all signals within a cluster are below their activation threshold. Assuming that the probability distribution of the collective activity in a cluster of size $K$ is a Boltzmann distribution and that the configuration of complete silence has a null energy, $P_{silence}(K)$ relates to the partition function of the distribution, i.e., $P_{silence}(K) = Z_K^{-1}$. Thus $\ln P_{silence}$ can be associated to an effective free energy $F(K) = -\ln Z_K$[34]. The effective free energy $F = \ln P_{silence}$ scales with the cluster size, i.e., $F \sim -K^{\tilde{\beta}}$, with an average exponent

$\langle \tilde{\beta} \rangle = 0.673 \pm 0.002$ across subjects (average relative estimation error: 2.27%; $\langle R_{EV} \rangle = 1.38$; Fig. 3b, e–h).

The last observable that we studied was the eigenvalue spectra of coarse-grained variables. For this, for each cluster of size $K$, we decomposed the covariance matrix into eigenvectors and studied the behavior of eigenvalues $\lambda$ as a function of their relative rank. Consistent with Meshulam et al.[33,34], we found that the spectra in clusters of different size $K$ collapsed when the rank was normalized, i.e., rank$/K$, and a power-law scaling of the eigenvalues as a function of their rank, i.e., $\lambda \sim (\text{rank}/K)^{-\mu}$, with an average exponent $\langle \mu \rangle = 0.328 \pm 0.001$ across subjects, followed by exponential truncation due to finite size effect (average relative estimation error: 1.88%; $\langle R_{EV} \rangle = 1.21$; Fig. 3c, e–h). The estimated least-squares exponent $\mu$ stabilized for $K > 8$ (Fig. 3d).

We further compared the measured PRG exponents to those obtained from shuffled data for which the correlations between

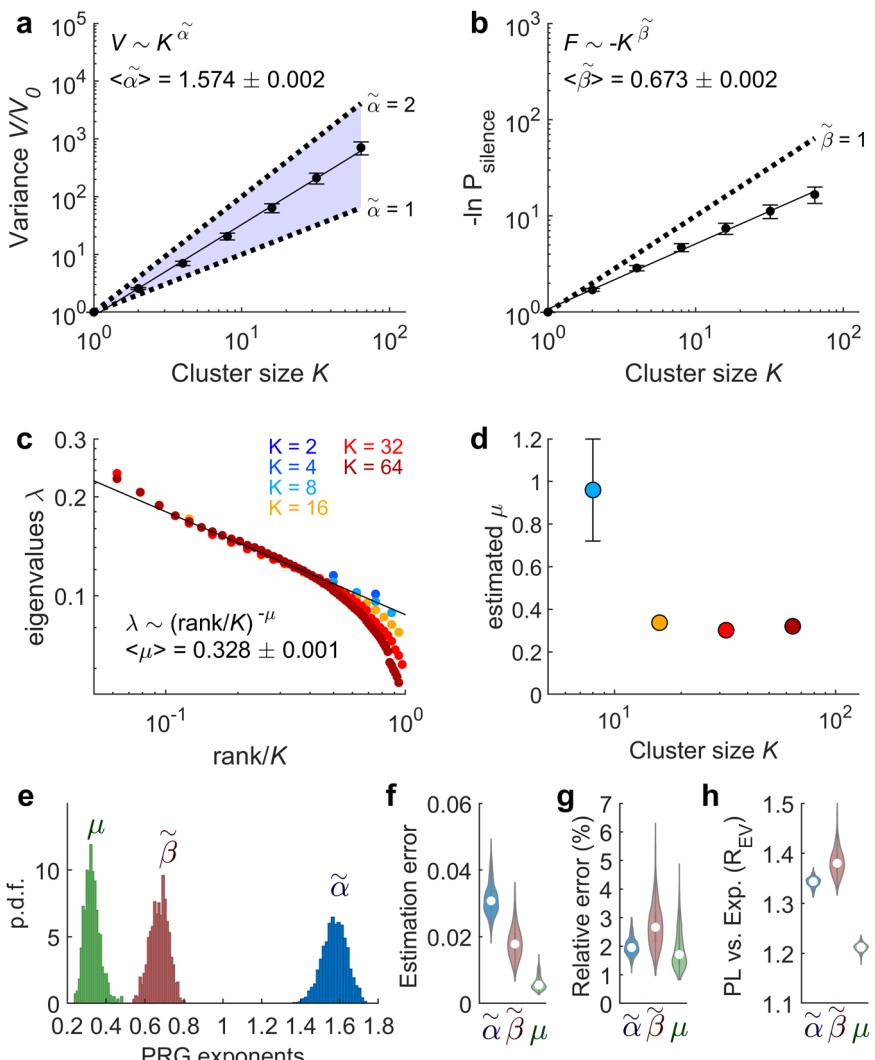

**Fig. 3 Phenomenological renormalization-group. a** Variance $V$ of coarse-grained variables as a function of cluster size $K$, average over subjects (black points; error bars indicate SD over subjects, $n = 1003$). The solid black line indicates least squares power law fit, i.e. $V = K^{\tilde{\alpha}}$. Dashed lines indicate linear ($\tilde{\alpha} = 1$) and quadratic ($\tilde{\alpha} = 2$) growths, corresponding to uncorrelated and fully correlated systems, respectively. $\langle \tilde{\alpha} \rangle$ indicates the average exponent across subjects. **b** Silence log-probability, $F = \ln P_{silence}$, of coarse-grained variables as a function of cluster size, average over subjects (black points; error bars indicate SD over subjects, $n = 1003$). The solid black line indicates least squares power law fit, i.e. $F = -K^{\tilde{\beta}}$. The dashed line indicates the prediction for uncorrelated variables ($\tilde{\beta} = 1$). In (**a**) and (**b**), the variance and the silence log-probability were normalized by their corresponding values at coarse-graining step $k = 0$ (original system). $\langle \tilde{\beta} \rangle$ indicates the average exponent across subjects. **c** Eigenvalues $\lambda$ of the covariance matrix as a function of their relative rank, for clusters of different sizes, for one example subject. The solid black line indicates least squares power law fit, i.e. $\lambda = \left(\frac{rank}{K}\right)^{-\mu}$, for $\frac{rank}{K} < 0.4$. $\langle \mu \rangle$ indicates the average exponent across subjects. **d** Estimated exponent $\mu$ for different cluster sizes. Error bars indicate the estimation error of the exponent (for $K > 8$ error bars are smaller than the symbols). **e** Distribution of exponents $\tilde{\alpha}, \tilde{\beta}, \mu$ for single-subject scans ($n = 1003$). **f** Least square estimation errors of PRG exponents. **g** Relative estimation error of exponents; e.g., $\Delta\tilde{\alpha}/\tilde{\alpha}$, where $\Delta\tilde{\alpha}$ is the least square estimation error of exponent $\tilde{\alpha}$ ($n = 1003$). White circles indicate medians. **h** The power-law fits of $V(K)$, $F(K)$, and $\lambda(rank/K)$ were compared to those obtained using an exponential function by calculating the ratio between the explained variance of the competing regression models ($R_{EV}$). Ratios >1 favor the power law hypothesis. Violin plots represent the distribution of ratios across subjects ($n = 1003$). White circles indicate medians.

signals were destroyed. To do this, the time frames of the binarized fMRI signals were randomly permuted, independently for each ROI. The PRG method applied to the shuffled data yielded exponents that were significantly different than those obtained using the original data (see Supplementary Fig. S2). Indeed, shuffled data yielded exponents $\tilde{\alpha}$ and $\tilde{\beta}$ that approach 1 (as expected for independent signals) and showed no evidence of scaling of the eigen-spectrum. Finally, we examined whether the number of ROIs affect the PRG scaling. For this, we built subsampled systems by randomly selecting a fraction of the $N$ ROIs and we applied the PRG method to the subsampled data. We found that PRG exponents from subsampled data converged

to those obtained using the full-size original data when the fraction of selected ROIs was larger than ∼ $0.7N$ (see Supplementary Fig. S3).

In conclusion, using the correlation function and the PRG method we were able to describe the scale invariance of collective fMRI binarized activity by means of four power-law exponents $\tilde{\eta}$, $\tilde{\alpha}, \tilde{\beta}$, and $\mu$.

**Connectivity-based phenomenological renormalization-group.** The above PRG approach was designed to study scale-invariance in neural systems when information about the neural connectivity

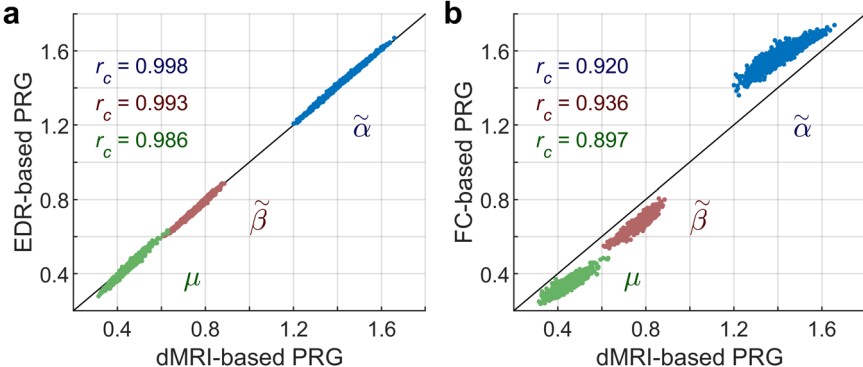

**Fig. 4 Connectivity-based phenomenological renormalization group. a** Connectivity-based PRG exponents calculated from the fMRI data assuming structural connections given by dMRI or EDR. Each dot represents an individual scan ($n = 1003$). $r_c$: correlation between corresponding exponents computed based on dMRI and EDR. **b** Comparison between dMRI-based PRG exponents and those obtained using the original PRG method (based on functional correlations, FC). Each dot represents an individual scan ($n = 1003$). $r_c$: correlation between corresponding exponents computed based on dMRI and FC.

was unknown[33]. This method is based on the correlation matrix (functional correlations) of variables at successive coarse-graining. We here extended this method to incorporate information about structural connectivity (see "Methods") that, in our case, was provided by the dMRI connectivity and its EDR approximation. For this, one needs to coarse-grain the variables and the connectivity at each coarse-graining step. When applying this connectivity-based PRG to the fMRI data, we found that $\tilde{\alpha}$, $\tilde{\beta}$, and $\mu$ exponents computed based on the dMRI connectivity strongly correlated with exponents computed based on the EDR (correlation > 0.98; Fig. 4a, see also Supplementary Fig. S4). Interestingly, the connectivity-based PRG exponents also strongly correlate with the exponents obtained using the original version of the PRG method, i.e., based on functional correlations (correlation > 0.89, Fig. 4b). In fact, we found that connectivity-based PRG exponents were shifted by a systematic bias from PRG exponents equal to −0.161, 0.095, and 0.106 for exponents $\tilde{\alpha}$, $\tilde{\beta}$, and $\mu$, respectively. We concluded that scaling exponents were consistently and reliably estimated using the PRG based on functional correlations or structural connections.

**Whole-brain spin model.** To test whether the observed scaling of activity is a signature of criticality, we built a spin model based on large-scale connectivity. The spin model is a canonical model presenting a second-order phase transition between ordered and disordered phases. Using the maximum entropy principle, the spin model can be mapped to binary neural data such as spiking activity (e.g. refs. [48,49]) or binarized fMRI data (e.g. refs. [31,46,50]). In this model, at each time step, the state of each node is described by a binary variable, i.e., $\sigma_i \in [-1, +1]$, and the collective activity of the $N$ nodes is given by a binary pattern $\vec{\sigma} = [\sigma_1, \ldots, \sigma_N]$. The probability of each pattern is given by the Boltzmann distribution:

$$P(\vec{\sigma}) = \frac{1}{Z} \exp\left( \beta \sum_{i,j} C_{ij} \sigma_i \sigma_j \right), \qquad (1)$$

Where $E(\vec{\sigma}) = -\beta \sum_{i,j} C_{ij} \sigma_i \sigma_j$ represents the energy of the pattern; $Z$ is the partition function, i.e., $Z = \sum_{\{\vec{\sigma}\}} \exp(-E(\vec{\sigma}))$; and $\beta$ is a scaling parameter of the connectivity matrix $C$, equivalent to an inverse temperature, i.e., $\beta = 1/T$, which is the free parameter of the model. The connectivity matrix $C$ was given by the EDR or the dMRI. Realizations of the spin model were obtained using Monte Carlo Metropolis simulations (see "Methods"). For each configuration $\vec{\sigma}$, the population activity is defined as the average

node value: $M(\vec{\sigma}) = \sum_i \sigma_i / N$. The average population activity $\langle M \rangle$ was obtained by averaging across simulation steps.

For temperatures lower than a critical value, $T_c = \beta_c^{-1}$, the spin model is in an ordered phase and presents a spontaneous population activity (i.e., $|\langle M \rangle| > 0$) that vanishes in the disordered phase for temperatures $> T_c$ (Fig. 5a). A power-law correlation function was observed around the critical point separating the two phases (Fig. 5b; see also Supplementary Fig. S5), with a critical exponent $\tilde{\eta}$ that was close to the one measured in the data for the EDR ($\tilde{\eta} = 0.515 \pm 0.013$; relative error from the empirical exponent: $\Delta_{\tilde{\eta}} = 0.4\%$), but slightly different for dMRI ($\tilde{\eta} = 0.310 \pm 0.009$; $\Delta_{\tilde{\eta}} = 39.6\%$). When fitting a power law to the model correlation function $g(r)$ for the full range of tested temperatures, we found that the power exponent was the closest to the empirical one around the critical point for the two connectivity matrices (Fig. 5c).

We next applied the PRG method to the model activity. We found that, for the EDR and close to the critical point, the scaling of the variance and the covariance eigen-spectrum was similar to the one measured in fMRI data: the critical exponents were equal to $\tilde{\alpha} = 1.62 \pm 0.01$ and $\mu = 0.30 \pm 0.03$ ($\Delta_{\tilde{\alpha}} = 2.9\%$, $\Delta_\mu = 8.5\%$; Fig. 5d–g). Power-law scaling of the coarse-grained variance was observed for all temperatures, while the eigen-spectrum scaled as a power law of the rank for the supercritical and critical regimes, but it exponentially decays in the subcritical regime (Supplementary Fig. S5). For the dMRI connectivity, the $\tilde{\alpha}$ critical exponent was similar to the one measured in the fMRI data, but the $\mu$ critical exponent deviated from the data: $\tilde{\alpha} = 1.53 \pm 0.05$ ($\Delta_{\tilde{\alpha}} = 2.9\%$) and $\mu = 0.26 \pm 0.02$ ($\Delta_\mu = 21.7\%$) (Fig. 5d–g). Thus, opposite to the case of the EDR, in the case of the dMRI connectivity the empirical exponents could not be simultaneously fitted using a unique temperature parameter. Similarly, when using the connectivity-based PRG to compare the scaling of the data and the model, we found that the model's critical exponents were remarkably close to the empirical exponents when the EDR was used both to couple and coarse-grain the spin variables ($\Delta_{\tilde{\alpha}} = 1.1\%$, $\Delta_\mu = 1.0\%$; see Supplementary Fig. S6c, d), but deviated for the dMRI connectivity ($\Delta_{\tilde{\alpha}} = 11.5\%$, $\Delta_\mu = 11.2\%$; see Supplementary Fig. S6a, b). We note that, since spin variables are symmetric (−1 or 1), we cannot define a silence probability to be associated to a free energy. Thus, the exponent $\tilde{\beta}$ could not be calculated for the model.

We concluded that, around its critical point, the spin model approximates the fMRI correlations and their scaling features, especially for the EDR, i.e., in the absence of long-range connections.

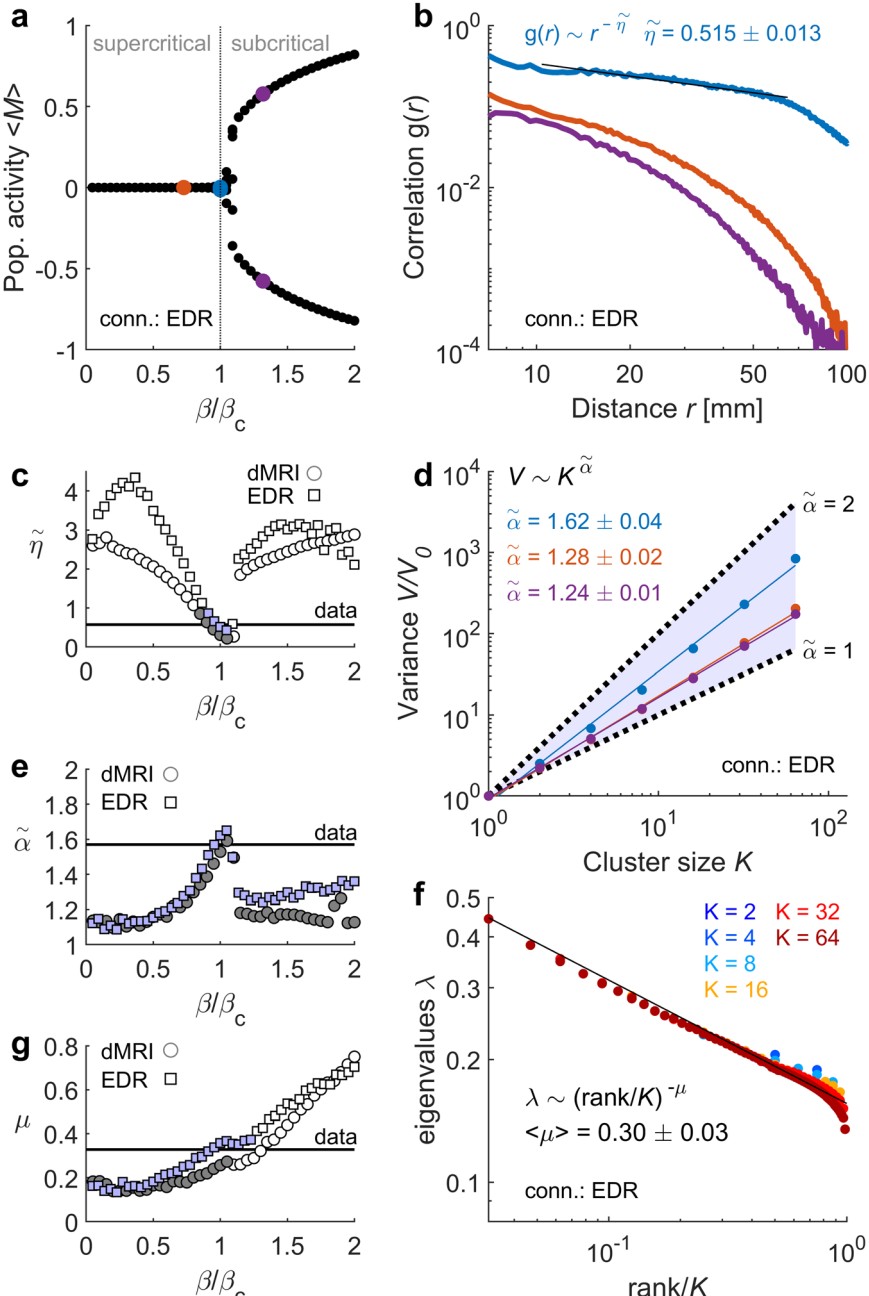

**Fig. 5 Spin model. a** Population activity as a function of $\beta = 1/T$, relative to the critical point $\beta_c = 1/T_c$. For $T > T_c$, the system is disordered and the average population activity is zero. For $T < T_c$, the system is ordered and a spontaneous population activity emerges and settles in either a negative or a positive value (depending on the initial conditions). **b** At the critical point (*blue*), the correlation function is a power law, with a power exponent $\tilde{\eta}$ close to the one measured in the fMRI data. The correlation function is shown for three example temperatures, also shown in (**a**) (*red*: supercritical; *blue*: critical; *purple*: subcritical). **c** Exponent $\tilde{\eta}$ as a function of $\beta/\beta_c$. **d** Variance $V$ of coarse-grained variables as a function of cluster size $K$, for the three example temperatures. **e** Exponent $\tilde{\alpha}$ as a function of $\beta/\beta_c$. **f** Eigenvalues of the covariance matrix as a function of their relative rank, at the critical point, for clusters of different sizes. **g** Exponent $\mu$ as a function of $\beta/\beta_c$. In (**a**), (**b**), (**d**) and (**f**), the connectivity used was the EDR. In (**c**), (**e**), and (**g**): filled symbols indicate explained variance ratios favoring the power law model over the exponential model, i.e., $R_{EV} > 1$ (see also Supplementary Fig. S5); exponent estimation errors are smaller than the symbols; the horizontal line indicates the empirically measured exponent.

## Discussion

We have shown that rs-fMRI signals present scaling features in the correlation function and as a function of coarse-graining based either on functional or structural connectivity. Notably, we found that the observed scaling can emerge from connections following a simple EDR and critical dynamics, i.e., the scaling exponents observed in the data were strikingly close to those predicted by a critical system of spins interacting through the EDR connectivity. Thus, our results suggest that criticality is the link between the connectome's structure and scale-invariant brain dynamics.

Previous theoretical work has tested the PRG method in an interacting particle system, the contact process, and the Ising model, both in a regular 2D lattice and with nearest-neighbor interactions[36]. The scaling of the contact process and Ising models under PRG yields critical exponents that differ between

3.4-27.2% from those measured from collective neuronal activity of mouse hippocampus, which was the target of those models[33,36]. Interestingly, critical exponents in the contact process were found to be unchanged in the presence of long-range interactions in small-world networks[36]. Our strategy, however, was different since we modeled the system with a proxy of the underlying brain connectivity, thus conserving both the size of the network and the spatial distribution of the nodes. This allowed us to map the scaling exponents observed in rs-fMRI data to the phases of a model presenting a phase-transition. In our case, using the EDR, the model's critical exponents were remarkably close to the exponents measured in rs-fMRI signals (both using the PRG or the connectivity-based PRG; see Fig. 5 and Supplementary Fig. S6), suggesting that brain dynamics operate close to a critical point in which order and disorder coexist. This result is in line with previous works showing that rs-fMRI signals display critical-like patterned activity detected using other methods, such as neuronal avalanches[9] and maximum entropy models which are equivalent to spin models inferred from data[31,50]. Since, the EDR has been previously observed in different species (rodents and nonhuman primates), it would be interesting to examine whether large-scale dynamics in these animal models also scale under PRG and relate to criticality.

Overall, our study shows that combination of PRG, connectomes, and models can be useful to distinguish the working regime of the observed neural system. Since different behavioral and pathological brain states deviate from critical dynamics[23–32], extending these analyses to different brain states could provide new insights on phase transitions in neural systems. We hypothesize that the PRG exponents change depending on the brain state and could be used as biomarkers in clinical and fundamental research. Indeed, in a recent study, Rocha et al.[29] have shown that criticality is lost in the case of stroke lesions, but it is recovered over time as behavior improves. Thus, criticality signatures can represent promising tools for translational research. Furthermore, we note that the PRG method can be formalized using different interaction measures (e.g., synchrony) and in Fourier space[33], which makes it suitable to study oscillatory dynamics recorded using LFPs, MEG, or electrocorticography. However, notice that, when applying the PRG method to finite data, both the cluster size's range on which scaling can be tested and the accuracy of coarse-grained statistics (e.g., covariances) are limited by the amount of data and the tradeoff between the spatial and temporal resolutions of the used recording technique.

The dMRI is a non-invasive method to estimate the large-scale brain connectivity, but it has methodological caveats and limitations[51]. In particular, it estimates short-range intra-hemispheric connections more reliably than inter-hemispheric ones. On the contrary, at the mesoscale, invasive chemical tracers are considered to be the gold standard for estimating the connecting fibers with high accuracy. This technique has shown that connection strength exponentially declines with distance. Here, we showed that the EDR with a characteristic scale that was consistent with the extrapolation to human brain given its volume[39], acheived a good linear prediction of fMRI signals, and yielded an accurate, consistent prediction of scaling exponents of brain activity using a critical whole-brain model. The whole-brain model constraint with the dMRI connectivity did not fit all scaling exponents neither at the critical point nor using a single temperature different from $T_c$ (see Fig. 5 and Supplementary Fig. S6). We concluded that, at least within the framework of our simple model, the EDR-based critical spin model represents a more parsimonious description of the observed rs-fMRI dynamics than the model based on dMRI connectivity (which presents long-range connections departing from the EDR). Future studies could explore the link between scaling features and

connectivity using different recording techniques and more realistic models including interacting excitatory and inhibitory neural populations.

In conclusion, we have shown that whole-brain dynamics display scaling properties that emerge from exponentially decaying connections and critical dynamics, which are two features of connectivity and dynamics largely supported by fiber-tracking research and studies of neural activity at different scales and with different techniques.

## Methods

**Neuroimaging ethics.** The Washington University–University of Minnesota (WU-Minn HCP) Consortium obtained full written informed consent from all participants to study procedures and data sharing outlined by HCP, and research procedures and ethical guidelines were followed in accordance with Washington University institutional review board approval.

**Functional MRI data.** In this study we analyzed publicly available rs-fMRI data from the Human Connectome Project (HCP), from 1003 participants. The participants were scanned on a 3 T connectome-Skyra scanner (Siemens). The rs-fMRI data was acquired for ~15 min, with eyes open and relaxed fixation on a projected bright cross-hair on a dark background. The HCP website (https://www.humanconnectome.org/) provides the details of participants, the acquisition protocol and preprocessing of the functional data. Briefly, the fMRI data was preprocessed using standardized methods using FSL (FMRIB Software Library), FreeSurfer, and the Connectome Workbench software[52,53]. This preprocessing included correction for spatial and gradient distortions and head motion, intensity normalization and bias field removal, registration to the T1 weighted structural image, transformation to the 2 mm Montreal Neurological Institute (MNI) space, using the FIX artefact removal procedure[53,54]. The head motion parameters were regressed out and structured artefacts were removed by ICA + FIX processing[55,56]. Preprocessed timeseries of all grayordinates are in HCP CIFTI grayordinates standard space and available in the surface-based CIFTI file for each participant. A custom-made Matlab script, using the ft_read_cifti function from the Fieldtrip toolbox[57], was used to extract the average timeseries of all the grayordinates in each region of the Schaefer parcellation, which are defined in the HCP CIFTI grayordinates standard space. Sequence and imaging parameters: Sequence: Gradient-echo EPI; TR: 720 ms; TE: 33.1 ms; flip angle 52 deg; FOV: 208 × 180 mm (RO x PE); Matrix: 104 × 90 (RO x PE); Slice thickness: 2.0 mm, 72 slices, 2.0 mm isotropic voxels; Multiband: factor 8; Echo spacing: 0.58 ms; BW: 2290 Hz/Px.

**Structural connectivity using dMRI.** Structural connectivity was estimated from diffusion spectrum and T2-weighted imaging data from 32 participants from the HCP database, scanned over 89 min. Acquisition parameters are described in detail in the HCP website[58]. The freely available Lead-DBS software package (http://www.lead-dbs.org/) provided the preprocessing which is described in detail in Horn and colleagues[59] but, briefly, the data was processed using a generalized q-sampling imaging algorithm implemented in DSI studio (http://dsi-studio.labsolver.org). Segmentation of the T2-weighted anatomical images produced a white-matter mask and co-registering the images to the b0 image of the diffusion data using SPM12. In each HCP participant, 200,000 fibers were sampled within the white-matter mask. Fibers were transformed into MNI space using Lead-DBS[60]. We used the standardized methods in Lead-DBS to produce the structural connectomes for the Schaefer 1000 parcellation scheme[61]. The connectivity weight $C_{ij} = C_{ji}$ was given by the number of fibers connecting two brain regions. To have values between 0 and 1, we normalized the weights by dividing them by the largest value, i.e., max($\boldsymbol{C}$). Diffusion MRI parameters: Sequence: Spin-echo EPI; TR: 5520 ms; TE: 89.5 ms; flip angle: 78 deg; refocusing flip angle: 160 deg; FOV: 210 × 180 (RO x PE); matrix: 168 × 144 (RO x PE); slice thickness: 1.25 mm, 111 slices, 1.25 mm isotropic voxels; Multiband factor: 3; Echo spacing: 0.78 ms; BW: 1488 Hz/Px; Phase partial Fourier: 6/8; b-values: 1000, 2000, and 3000 s/mm2.

**Schaefer parcellation.** Schaefer and colleagues created a publicly available population atlas of cerebral cortical parcellation based on estimation from a large data set ($n = 1489$)[61]. They provide parcellations of 400, 600, 800, and 1000 areas available in surface spaces, as well as MNI152 volumetric space. We used here the Schaefer parcellation with 1000 areas and estimated the Euclidean distances from the MNI152 volumetric space and extracted the timeseries from HCP using the HCP surface space version.

**Data binarization.** The rs-fMRI time-series were binarized to study the data statistics and to compared them to those predicted by the spin model. For each scan, the z-scored time-series of each ROI, $z_i(t)$ ($1 \le i \le N$), was binarized by imposing a threshold $\theta = 1$. The binarized activity was $\sigma_i(t) = 1$ if $z_i(t) > \theta$ and $\sigma_i(t) = 0$ otherwise. Transformation of continuous signals into discrete point processes has proven to effectively capture and compress fMRI large-scale dynamics[9].

Importantly, the fluctuations that cross the threshold do not merely represent noise, since the resulting point process largely overlaps with deconvoluted fMRI signals using the hemodynamic response function and preserves the topology of the resting state networks[9]. Furthermore, using maximum entropy models to estimate the probability distribution of binarized activity, it has been shown that binarized rs-fMRI data is poised close at a critical point[31].

**Correlation function**. We calculated the average correlation $g(r)$ as a function of the Euclidean distance $r$ between ROIs. For this, we calculated the average correlation among pairs of nodes that were separated by distances between $r$ and $r + \Delta r$, with $\Delta r = 0.43$ mm, i.e.:

$$g(r) = \frac{1}{N_r} \sum_{r_{ij} \in [r, r+\Delta r]} c_{ij}, \tag{2}$$

where $N_r$ is the number of pairs of ROIs $(i, j)$ such that $r_{ij} \in [r, r + \Delta r]$, and $r_{ij}$ and $c_{ij}$ denote the distance and the Pearson correlation between ROIs $i$ and $j$, respectively. Distances between ROIs range between 4.28 mm and 173.16 mm.

**Phenomenological renormalization-group method**. We here review the recently proposed PRG approach to study scale-invariance in neural systems[33]. Within this method, the collective activity is iteratively coarse-grained by grouping together the variables that are maximally correlated.

Let $\sigma_i^{(0)}$ be the binary activity of ROI $i$ for $i = 1, \ldots, N$, with $\sigma_i^{(0)} \in \{0, 1\}$. The superscript 0 indicates that the data is not coarse-grained. In the first coarse-graining step, we seek for the pair of variables $\{i^*, j^*\}$ with maximal correlation and sum them:

$$\sigma_{i'}^{(1)} = \sigma_{i^*}^{(0)} + \sigma_{j^*}^{(0)}, \tag{3}$$

where $i' = 1, \ldots, N/2$. We repeat this procedure for the second maximally correlated pair among the remaining variables, i.e., from the set $\{i, j \in \{1, \ldots, N\} : i, j \notin \{i^*, j^*\}\}$, and so on until all pairs are used. This process is iterated for coarse-grained variables $\sigma_i^{(k)}$, resulting in clusters of size $K = 2, 4, \ldots, 2^k$. The size of the system is equal to $N_k = N/(2^k)$ at each coarse-graining step.

Along the coarse-graining procedure, several statistics of $\sigma_i^{(k)}$ are calculated and their change at different coarse-graining steps are examined. A first observable is the variance of coarse-grained variables:

$$V(K) = \frac{1}{N_k} \sum_{i=1}^{N_k} \left\langle \left( \sigma_i^{(k)} \right)^2 \right\rangle - \left\langle \sigma_i^{(k)} \right\rangle^2. \tag{4}$$

For calcium imaging recordings in the mouse hippocampus, it has been shown that the variance scales with the cluster size, $V \propto K^{\tilde{\alpha}}$[33], with a power-law exponent that lies between linear ($\tilde{\alpha} = 1$) and quadratic growth ($\tilde{\alpha} = 2$), corresponding to uncorrelated and full-correlated systems, respectively.

A second quantity is the probability of silence, $P_{\text{silence}}(K) = P(\sigma_i^{(k)} = 0)$ for all $i$. Assuming that the probability distribution $P_K(\vec{\sigma}^{(k)})$ of the collective activity in a cluster of size $K$ is a Boltzmann distribution and that the configuration of complete silence has null energy, $P_{\text{silence}}(K)$ relates to the partition function of the distribution, i.e., $P_{\text{silence}}(K) = Z_K^{-1}$. Thus $\ln P_{\text{silence}}(K)$ can be associated to a free energy $F(K) = -\ln Z_K$. In calcium imaging recordings in the mouse hippocampus, it has been shown that the free energy scales with the cluster size as $F \propto -K^{\tilde{\beta}}$[34], with an exponent <1, which is the expected value for independent variables.

A third quantity is the spectrum of the covariance matrix inside a cluster of size $K$. Let $\lambda$ denote the eigenvalues of the covariance matrix. The eigenvalues are ordered from the highest eigenvalue, rank = 1, to the lowest, rank = $K$. It has been shown that, at the fixed point of RG, one expects that the eigen-spectrum scales with the relative rank[33]:

$$\lambda \propto \left( \frac{\text{rank}}{K} \right)^{-\mu}. \tag{5}$$

Notice that the eigen-spectrum presents scaling in two senses: the spectra in clusters of different size $K$ collapse when the rank is normalized, i.e., rank/$K$, and the eigenvalues have a power-law decay as a function of rank, followed by exponential truncation due to finite size effect[33].

Spectral properties of covariance matrices often depend on the ratio between the number of samples and the number of variables. For this reason and following Meshulam et al.[34], we studied the eigen-spectrum for cluster sizes for which we have >10 times more samples than variables. In our case, with 1200 time frames per scan we required $K \leq 2^6 = 64$. Throughout this article, we coarse-grained the activity up to six times and evaluated the PRG power-laws of $V(K)$ and $F(K)$ for $1 \leq K \leq 64$. The eigen-spectrum was computed for $K \leq 64$ and its power-law fit was evaluated for $K^{-1} < \text{rank}/K < 0.4$ to avoid the finite-size truncation[36].

In a $D$-dimensional system with translational invariance and a power-law correlation function $g(r) \sim r^{-\tilde{\eta}}$ (as expected for a critical system), the exponent $\mu$ is related to $\tilde{\eta}$. In this case, the eigenvalues are given by the Fourier transform of the correlation function[62]:

$$\lambda\left(\vec{k}\right) = \int d^D r \, g(r) e^{i\vec{k}.\vec{r}} \sim \frac{1}{\left|\vec{k}\right|^{D-\tilde{\eta}}}. \tag{6}$$

Since the rank scales as rank $\sim \left|\vec{k}\right|^D$, one has: $\mu = (D - \tilde{\eta})/D$. However, the $\tilde{\eta}$ exponent does not satisfy this relation neither in the critical spin model nor in the fMRI data. This might indicate that the system is not translational invariant.

**Connectivity-based phenomenological renormalization-group method**. Given that, in our case, we have information about the structural connectivity, we can extend the PRG method to coarse-grain the collective fMRI activity based on this structural connectivity matrix $\mathbf{C}$. Here, the matrix $\mathbf{C}$ is given by the dMRI or the EDR. Note that in both cases, the connectivity matrix is symmetric. In the connectivity-based PRG (CBPRG) method, we grouped the variables that are maximally connected. In the first coarse-graining step, we seek for the pair of variables $\{i^*, j^*\}$ with maximal connection $C_{i^* j^*}^{(0)}$, where $\mathbf{C}^{(0)} = \mathbf{C}$, and sum them as in Equation (3). This was repeated for the second maximally connected pair among the remaining variables and so on until all pairs are used. Let $m'$ and $n'$ be two indices corresponding to two groups formed in the first step from variables $\{i, j\}$ and $\{k, l\}$, i.e.:

$$\sigma_{m'}^{(1)} = \sigma_i^{(0)} + \sigma_j^{(0)}, \tag{7}$$

$$\sigma_{n'}^{(1)} = \sigma_k^{(0)} + \sigma_l^{(0)}, \tag{8}$$

Where $i, j, k, l \in \{1, \ldots, N\}$ and $m', n' \in \{1, \ldots, N/2\}$. A new connectivity matrix of size $(N/2) \times (N/2)$ is defined as follows:

$$C_{m'n'}^{(1)} = \frac{1}{4}\left[ C_{ik}^{(0)} + C_{il}^{(0)} + C_{jk}^{(0)} + C_{jl}^{(0)} \right]. \tag{9}$$

This connectivity matrix $\mathbf{C}^{(1)}$ was then used to group the variables at step 2. As above, the process was iterated to obtain coarse-grained variables $\sigma_i^{(k)}$, built by grouping variables $\sigma_i^{(k-1)}$ based on their connectivity $\mathbf{C}^{(k-1)}$, resulting in clusters of size $K = 2, 4, \ldots, 2^k$. Notice that the weights of matrix $\mathbf{C}^{(k)}$ result from averaging $4^k$ connectivity weights from the original connectivity matrix.

**Whole-brain spin model**. To relate the observed fMRI statistics to critical dynamics, we built a spin model based on large-scale connectivity. In this model, the state of each node is described by a binary variable or "spin", i.e., $\sigma_i \in [-1, +1]$, and the collective activity of the $N$ nodes is given by a binary pattern or configuration $\vec{\sigma} = [\sigma_1, \ldots, \sigma_N]$. The probability of each pattern is given by the Boltzmann distribution:

$$P(\vec{\sigma}) = \frac{1}{Z} \exp[-E(\vec{\sigma})], \tag{10}$$

where $E(\vec{\sigma})$ represents the energy of the pattern and is given as:

$$E(\vec{\sigma}) = -\beta \sum_{i,j} C_{ij} \sigma_i \sigma_j. \tag{11}$$

$Z$ is the partition function, i.e., $Z = \sum_{\{\vec{\sigma}\}} \exp(-E(\vec{\sigma}))$. Spins interact through the connectivity matrix $\mathbf{C}$. $\beta$ is a scaling parameter of the connectivity matrix $\mathbf{C}$; it is equivalent to an inverse temperature, i.e., $\beta = 1/T$, which is the free parameter of the model.

For each configuration $\vec{\sigma}$, the population activity is defined as the average node value: $M(\vec{\sigma}) = \sum_i \sigma_i / N$. The average population activity $\langle M \rangle$ was obtained by averaging across simulation steps. The model presents a second-order phase transition that can be detected by examining the behavior of $\langle M \rangle$ as a function of the temperature parameter. For temperatures lower than a critical value, $T_c = \beta_c^{-1}$, the spin model presents a spontaneous population activity, i.e., $|\langle M \rangle| > 0$ (subcritical regime), that vanishes for temperatures $> T_c$ (supercritical regime).

Realizations of the spin model were obtained using Monte Carlo Metropolis simulations. The algorithm starts with an initial random configuration of $N$ spins, then flips the spin of a randomly chosen node, and calculates the change in energy $\Delta E$ induced by the spin flip. If $\Delta E < 0$, the spin flip is accepted, otherwise it is accepted with probability $\exp(-\beta \Delta E)$. For each tested value of $\beta$, we ran 5 realizations (with different initial conditions) of $50,000 \times N$ simulation steps. The system's configuration was stored every $N$ flip attempts. To avoid dependences on initial conditions, simulations started with extra 500,000 steps without storing the configurations. The population activity, the correlation function, the coarse-grained variance, and the coarse-grained eigen-spectrum were averaged over realizations, for each $\beta$.

Note that, using the maximum entropy principle, the spin model (and its extensions) can be mathematically map to binary data. This has been done using spiking data at the microcircuit level (e.g. refs. [48,49]) but also using binarized fMRI data at the large-scale level (e.g. refs. [31,46,50]). Briefly, within the framework of maximum entropy models, to estimate the probability distribution $P(\vec{\sigma})$ of binary patterns, one seeks for the distribution that matches some statistics of the data and

has maximum entropy. It is known that the maximum entropy distribution that preserves the pairwise correlations of the data is $P(\vec{\sigma}) = \frac{1}{Z}\exp(\sum_{i,i}J_{ij}\sigma_i\sigma_j)$, where $J_{ij}$ is the effective connectivity between the variables $\sigma_i$ and $\sigma_j$[49]. The resulting maximum entropy distribution is thus equivalent to the Boltzmann distribution of the present spin model.

**Power law fit.** We fitted power laws using least squares on log-log scattered data. Note that we were unable to use a maximum likelihood estimation (MLE) approach which applies only to probability distributions. MLE, together with Kolmogorov–Smirnov (KS) statistics[63,64] or log-likelihood ratios (LLR) between the candidate heavy-tailed distributions[65], are commonly used to study critical power-law behavior in neural systems[12,66,67]. However, this approach is only applicable to probability densities. For this reason, we here used least squares to fit power laws and we evaluated the goodness of fit by comparing the explained variance of the least-squares fit using the power law and the one obtained using an exponential function. Specifically, we calculated the ratio $R_{EV} = R_{PL}^2/R_{Exp}^2$, where $R_{PL}^2$ is the explained variance (or coefficient of determination) of the linear regression model $\log Y = a \log X + b$ (power law) and $R_{EV}^2$ is the explained variance of the linear regression model $\log Y = cX + d$ (exponential). Ratios $R_{EV} > 1$ favor the power law hypothesis against the exponential alternative. The estimation error $\Delta a$ of the power-law exponent was given by the error of the slope $a$ of the linear regression model $\log Y = a \log X + b$. The relative error was defined as: $100 \times \Delta a/a$.

**Statistics and reproducibility.** The goodness of the linear prediction $X_{\text{pred}} = CX$ was given by the Pearson correlation between $X_{\text{pred}}(t)$ and $X(t)$ for all nodes and all subjects ($n = 1000 \times 1003$), where $C$ denotes the dMRI or the EDR connectivity matrices and $X$ denotes the data signals (Fig. 1d). The distributions of correlation coefficients were compared to the one obtained using a shuffled connectivity. These comparisons were done using a Welch's $t$-test, after Fisher z-transformation of the correlation coefficients.

For each subject, shuffled data were built by randomly permuting the time frames of binarized fMRI signals for each ROI separately. Next, the PRG method was applied to the resulting shuffled data (Supplementary Fig. S2). The distributions of the PRG exponents $\tilde{\alpha}, \tilde{\beta}, \mu$ from the original data and the shuffled data were compared using Wilcoxon tests ($n = 1003$ for each distribution).

MATLAB (R2021a) software was used to perform all analyses and to simulated the model. Numerical simulations were performed in a 50-nodes computer cluster (Intel® Xeon® E5-2684 at 2.1 Ghz, 256 GB RAM, 1 TB disk).

**Reporting summary.** Further information on research design is available in the Nature Portfolio Reporting Summary linked to this article.

## Data availability

We used a publicly available dataset of fMRI data from the Human Connectome Project (HCP), from 1003 participants selected from the March 2017 public data release from the Human Connectome Project (HCP). The HCP dataset is available at https://www.humanconnectome.org/study/hcp-young-adult. Source data for main figures presented in this study are provided as Supplementary Data 1–5.

## Code availability

The codes to perform the PRG analysis and to simulate the model are available at https://github.com/adrianponce/Scaling-of-whole-brain-resting-state-dynamics, https://doi.org/10.5281/zenodo.7962109[68].

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

## Acknowledgements

A.P.-A. was supported by a Ramón y Cajal fellowship (RYC2020-029117-I) from FSE/Agencia Estatal de Investigación (AEI), Spanish Ministry of Science and Innovation. A.P.-A. and G.D. were supported by the EU Fet Flagship Human Brain Project SGA3 (945539). G.D. was supported by the Spanish Research Project AWAKENING (PID2019-105772GB-I00/AEI/10.13039/501100011033), financed by the Spanish Ministry of Science, Innovation and Universities (MCIU), State Research Agency (AEI). M.L.K. is supported by the Centre for Eudaimonia and Human Flourishing (funded by the Pettit and Carlsberg Foundations) and Center for Music in the Brain (funded by the Danish National Research Foundation, DNRF117).

## Author contributions

A.P.-A. and G.D. designed the research. A.P.-A., M.L.K. and G.D. analyzed/processed data. A.P.-A. designed and performed model analysis and wrote the manuscript.

## Competing interests

The authors declare no competing interests.
