## [Peer Review File · Communications Biology]

Reviewers' comments:

Reviewer #1 (Remarks to the Author):

The authors analyse a dataset including human resting state fMRI data and diffusion MRI (dMRI) data. Firstly, they compare the structural connectivity derived from dMRI with a more schematic connectivity that decays exponentially with distance (EDR). More specifically they compare the probability distribution of the correlation between the recorded time-series and the time-series predicted via a linear model and report that the Pearson correlation coefficients calculated using EDR and dMRI connectivity matrices are remarkably similar.

Next, they measure the pairwise correlation between ROIs in the functional data as a function of (Euclidean) distance and they show that a power-law fits the data better than an exponential function. They employ a recent technique based on the concepts of renormalization group and report evidences of scale invariance in the fMRI data, similar to the ones showed in the original paper by Meshulam et al, for data of calcium imaging in CA1.

Lastly, they apply the phenomenological renormalization group approach to an Ising model where the connectivities are given by dMRI or EDR ("whole-brain spin model"). They compare the scaling exponents that they find at the critical point of the model with the exponents found for the fMRI data.

The main question that the authors address, i.e. how scale invariance emerges from neural interactions is certainly of interest. However, I find that the analyses reported in the paper are and not fully convincing.

In general, error bars are not reported in most of the panels. The authors mostly report error intervals for the exponents of the fitted power law as least square errors of the linear fit in a log-log plot, but they do not consider partitions over random quarters of the data (as in the original work), or variability between different recording sessions/individuals. I am worried that these factors may generate the very small error bars in Fig.3D for example. I am also concerned by the lack of controls in the results shown here. How do the exponents behave if data are reshuffled across times and/or ROIs?

1. Fitting data to power laws is always a controversial issue; see for instance,

Clauset, A., Shalizi, C. R., and Newman, M. E. (2009). Power-law distributions in empirical data. *SIAM review*, 51(4), 661-703;

A. Deluca and A. Corral. Fitting and goodness-of-fit test of non-truncated and truncated power-law distributions. *Acta Geophys.*, 61:13511394, 2013.

This work relies extensively on the fit to power laws and the the numerical values of the exponents when making claims about the dynamical regime of the system but not enough rigor is used in the evaluation of the exponents.

For example the measure of the power-law exponent is a function of the interval chosen, as shown in Fig.2D,E for r_{\min} and r_{\max} . How has this interval been chosen in the rest of the figures? More precision on this aspect is necessary.

2. When the authors measure the functional correlations as a function of ROI distance (Fig.2D,E), they report a power law fit on very small intervals of distances (e.g 40-60mm), whose meaning is debatable to say the least. I am also generally confused by this figure: if I look at panel A, the fit in

the interval [40,100] mm seems to give a smaller exponent (~ 0.4) than the one reported in panel D (~ 0.6). Conversely, panel A indicates a fit over an interval $r_{\min}=10\text{mm}$, $r_{\max}=90\text{mm}$ that is not included in the panels below.

3. Do the measured exponents depend on n_f/N ? The spectral properties of covariance matrices often depend on the ratio of the numbers of data and the number of variables. For example what happens when the analysis is repeated with subsampled ROIs? More analyses/discussions on this aspect are needed.

4. As mentioned by the authors, the PRG approach was proposed to obviate to the lack of knowledge of the topology of the network. The correlation structure of the data is then considered as a proxy for spatial proximity.

In this case, however, the connectivity structure is measured through dMRI (and simplified to EDR). Can the authors perform coarse-graining by grouping ROIs according to structural information? It could also be interesting to compare the clusters resulting from the different clustering rules, to further validate the PRG method.

5. I do not fully understand the meaning of comparing the empirical results to the PRG on the Ising model (comparison seems to be merely based on the fact that Ising model has a second order phase transition). For example what does the temperature represent for a whole-brain model? Why is the Ising model chosen rather than a different one? In principle one could choose a different model showing a second order phase transition, like the contact process. Then the exponents at the critical point would presumably be different and could for example be more similar to the ones measured in the data for dMRI (instead of EDR)?

6. In Figure 4 the behavior of the model is shown for the critical point as well as at a subcritical and a supercritical point. Several quantities are measured in all 3 regimes and fitted with power laws, however their expected behavior is not power law away from the critical point!

Also, in Fig.4D the authors find non-trivial scaling away from the critical point (and a very similar $\tilde{\alpha}$ exponent for the subcritical and the supercritical regimes). This is somewhat surprising and would deserve at least a short discussion.

Minor points:

A. The claim that dMRI connectivity "can be approximated" as an exponentially decaying function is bold and not fully supported by the evidence provided. The authors argue that dMRI data points toward a connectivity matrix where the strength of the interactions decays with (Euclidian) distance between ROIs as a power law, due to long-range (inter-hemispheric) interactions. On the other hand, exponentially decaying connectivities are essentially different network structures, only showing local interactions. I think that the fact that PRG exponents are similar for EDR and dMRI and the fact that a linear model predicts the times series almost equally well for the two structures does not convincingly support the claim that dMRI connectivity "can be approximated" as an exponentially decaying function. I suggest to soften the claim or provide stronger evidence.

B. The fit in figure 4F is hard to see due to the fact that the span of the y axis is much larger than the span of the curve.

C. Line 77: the authors claim that the phenomenological renormalization group approach "remains to be tested [...] using theoretical models". However, as they also point out in the discussion, the method

has been applied and its limits of validity have been discussed for the contact process and the Ising models (Nicoletti et al. 2020, cited).

D. I find the use of the expression "geometry of the brain" in the abstract quite confusing.

Reviewer #2 (Remarks to the Author):

The paper analyzes resting state fMRI dynamics based on the phenomenological renormalization group method and reports that brain dynamics display power-law correlations. An Ising-type model is used to further distinguish the fMRI results from noise or shuffled data. The paper is clear and well-written, but there seem to be a few points that need to be answered.

1.) The reported power laws are rarely spanning a full decade, although the typical requirement for the acceptance of a power-law interpretation of the data is two orders of magnitude. E.g. scale invariance within less than an order of magnitude is difficult to grasp. I understand that a statistically more solid work would presuppose an unrealistic amount of 1M recordings, which seems an inherent limitation of the empirical renormalization group.

2.) Both the model (Bialek, Berry and others) and the data analysis (e.g. Hilgetag, Kaiser and others) are not without earlier result of a similar type, which requires some discussion.

3.) The paper reports criticality, but does not discuss why this should be expected (or unexpected). 20 years after Beggs and Plenz paper, it seems legitimate to ask about the function and relevant at the particular modality where the data have been recorded.

Please check the last sentence of the abstract: Validation in a general sense is not possible in a way as described. Also, the "geometry of the brain" is not enough discussed in the paper to be mentioned here.

Fig 1D: Please find a better way to indicated correspondence between model and data, as the colors are not unambiguous (or there is a problem with my color vision).

Fig 1E: The error bar appear to indicate the standard deviation rather than SEM, but in either case the extraction of a nonlinear relation from these data is questionable.

163: Explained variance does not seem to be appropriate to justify a power-law, as the power-law may not even have a variance or the data variance in could be mainly due to the region below some lower cut-off.

Fig 4B: There is little reason to believe that the curves are straight lines. Instead, it seems to be a subcritical case, where the estimated power-law exponents do not provide additional information. This applies least for the red and the purple curve, whereas the blue curve, which may be the critical case, needs to represented differently, in its present form no judgement is possible, which is also unacceptable.

Fig 4F: It should be possible to change the scale of the y-axis so that the power-law relation becomes more obvious.

Reviewers' comments:

Reviewer #1 (Remarks to the Author):

The authors analyse a dataset including human resting state fMRI data and diffusion MRI (dMRI) data. Firstly, they compare the structural connectivity derived from dMRI with a more schematic connectivity that decays exponentially with distance (EDR). More specifically they compare the probability distribution of the correlation between the recorded time-series and the time-series predicted via a linear model and report that the Pearson correlation coefficients calculated using EDR and dMRI connectivity matrices are remarkably similar. Next, they measure the pairwise correlation between ROIs in the functional data as a function of (Euclidean) distance and they show that a power-law fits the data better than an exponential function. They employ a recent technique based on the concepts of renormalization group and report evidences of scale invariance in the fMRI data, similar to the ones showed in the original paper by Meshulam et al, for data of calcium imaging in CA1. Lastly, they apply the phenomenological renormalization group approach to an Ising model where the connectivities are given by dMRI or EDR (“whole-brain spin model”). They compare the scaling exponents that they find at the critical point of the model with the exponents found for the fMRI data.

The main question that the authors address, i.e. how scale invariance emerges from neural interactions is certainly of interest. However, I find that the analyses reported in the paper are and not fully convincing.

1. In general, error bars are not reported in most of the panels. The authors mostly report error intervals for the exponents of the fitted power law as least square errors of the linear fit in a log-log plot, but they do not consider partitions over random quarters of the data (as in the original work), or variability between different recording sessions/individuals. I am worried that these factors may generate the very small error bars in Fig.3D for example. I am also concerned by the lack of controls in the results shown here. How do the exponents behave if data are reshuffled across times and/or ROIs?

► We now indicate in all figure legends what error bars represent. We didn't use quarters as suggested by the reviewer because with random partitions of $\frac{1}{4}$ of the data the errors are still smaller than the symbols of the graphs (this is the case even for partitions of $\frac{1}{100}$ of the data, corresponding to groups of 10 subjects). Instead, we now show the standard deviation across subjects; see Figs. 3A and 3B. Also, we now report the distribution of PRG exponents across individual scans in the new panel Fig. 3E. We also report the distribution of the uncertainty (relative estimation error) of $\tilde{\eta}$ and PRG exponents in the new panels Fig. 2C and Fig. 3F. Note that, on average, the uncertainty of exponents was $< 3\%$. See lines 192-196, 204-205, and 233-234.

Furthermore, following the reviewer's recommendation, we also computed the PRG exponents on data that were reshuffled across time frames. The results are now shown in the new **Supplementary Fig. S1** and described in lines 236-242 in the main text. We found that the PRG method applied to the shuffled data yielded exponents that were significantly different than those obtained using the original data.

2. Fitting data to power laws is always a controversial issue; see for instance,

Clauset, A., Shalizi, C. R., and Newman, M. E. (2009). Power-law distributions in empirical data. SIAM review, 51(4), 661-703;

A. Deluca and A. Corral. Fitting and goodness-of-fit test of non-truncated and truncated power-law distributions. *Acta Geophys.*, 61:13511394, 2013.

This work relies extensively on the fit to power laws and the numerical values of the exponents when making claims about the dynamical regime of the system but not enough rigor is used in the evaluation of the exponents.

► Note that, in our study, we fitted power laws to scatter data that did not represent probability distributions. As mentioned by the reviewer, a maximum likelihood estimation (MLE) approach is recommended in the case of probability distributions. MLE, together with Kolmogorov–Smirnov (KS) statistics (Clauset et al., 2009; Deluca and Corral, 2013) or log-likelihood ratios (LLR) between the candidate heavy-tailed distributions (Alstott et al., 2014), are commonly used to study critical power-law behavior in neural systems (Marshall et al., 2016; Ponce-Alvarez et al., 2018; Yu, 2022). However, this approach is only applicable to probability densities. For this reason, we used least squares to fit power laws (i.e., linear fits on log-log scale as in Meshulam et al. 2018, 2019) and we evaluated the goodness of fit by comparing the explained variance using the power law and the one obtained using an exponential function (ratio R_{EV}). In the new version of the manuscript, we added a justification and a description of the power-law fitting procedure in the Methods section; see lines 602-617. We also added the distribution of R_{EV} for the PRG exponents in the new panel **3G** for the original data and for the controls asked by the reviewer (see below).

3. For example the measure of the power-law exponent is a function of the interval chosen, as shown in Fig.2D,E for r_{min} and r_{max} . How has this interval been chosen in the rest of the figures? More precision on this aspect is necessary.

► We now added a description of intervals chosen to fit the power laws of the PRG method; see lines 530-536. As mentioned in the response to point 3 raised by the reviewer, spectral properties of covariance matrices often depend on the ratio between the number of samples and the number of variables. For this reason and following Meshulam et al. (2018), we studied the eigen-spectrum for cluster sizes for which we have > 10 times more samples than variables. In our case, with 1,200 time frames per scan we required $K \leq 2^6 = 64$ (a further coarse-graining step would lead to $2^7 = 128$ variables in each cluster, i.e., a number of samples < 10 times larger than the number of variables). We thus coarse-grained the activity up to six times and evaluated the PRG power-laws of $V(K)$ and $F(K)$ for $1 \leq K \leq 64$. The eigen-spectrum was computed for $1 \leq K \leq 64$ and its power-law fit was evaluated for $\text{rank}/K < 0.4$ (similar to Meshulam et al., 2018, 2019) so that the finite-size decay for large relative ranks is not considered. See lines 526-536.

4. When the authors measure the functional correlations as a function of ROI distance (Fig.2D,E), they report a power law fit on very small intervals of distances (e.g 40-60mm), whose meaning is debatable to say the least. I am also generally confused by this figure: if I look at panel A, the fit in the interval [40,100] mm seems to give a smaller exponent (~ 0.4) than the one reported in panel D (~ 0.6). Conversely, panel A indicates a fit over an interval $r_{min}=10\text{mm}$, $r_{max}=90\text{mm}$ that is not included in the panels below.

► We indeed did a mistake in the plots of panels 2D and 2E in the previous manuscript version. We now have fixed it. Note that these panels have become panels 2E and 2F in the new manuscript version. We thank the reviewer for noticing the error and we apologize for it.

5. Do the measured exponents depend on n_f/N ? The spectral properties of covariance matrices often depend on the ratio of the numbers of data and the number of variables. For example what happens when the analysis is repeated with subsampled ROIs? More analyses/discussions on this aspect are needed.

► Following the reviewer's recommendation, we applied the PRG to subsampled data. For this, we built subsampled systems by randomly selecting a fraction of the N ROIs. We found that the PRG exponents from subsampled data converged to those obtained using the full-size original data when the fraction of selected ROIs was larger than $\sim 0.7N$. The results are now shown in **Supplementary Fig. S2** and described in lines 242-246.

As mentioned above, the reviewer is right: spectral properties of covariance matrices often depend on the ratio between the number of samples and the number of variables. For this reason, we followed the recommendation of Meshulam et al. and used $K \leq 2^6 = 64$ to obtain cluster sizes for which we have > 10 times more samples than variables (see lines 526-536).

6. As mentioned by the authors, the PRG approach was proposed to obviate to the lack of knowledge of the topology of the network. The correlation structure of the data is then considered as a proxy for spatial proximity. In this case, however, the connectivity structure is measured through dMRI (and simplified to EDR). Can the authors perform coarse-graining by grouping ROIs according to structural information? It could also be interesting to compare the clusters resulting from the different clustering rules, to further validate the PRG method.

► This is an excellent suggestion. Following the reviewer's recommendation, we extended the PRG to coarse-grain the variables based on structural connections. The method is now described in lines 545-562 in the main text. Note that we need to coarse-grain the connectivity at each coarse-graining step. Interestingly, this connectivity-based PRG yields exponents that strongly correlate with the exponents obtained using the original version of the PRG. Moreover, using the dMRI or the EDR connectivity to coarse-grain the variables yield practically the same results. These results are now shown in the new **Fig. 4** and **Supplementary Fig. S3**, and they are described in lines 251-266 in the main text. We also test the connectivity-based PRG on the model. We found that this version of the analysis also indicates that the ERD-constraint spin model best approximates the scaling exponents of fMRI activity exactly at the critical point; see lines 314-319 in the main text and the new **Supplementary Fig. S5**. When the model was built using the dMRI connectivity, the empirical exponents could not be simultaneously fitted using a unique temperature parameter (see **Supplementary Fig. S5**). We also refer to this analysis in the abstract (see line 28) and in the Discussion (see lines 342 and 358-359). We thank the reviewer for this valuable suggestion.

7. I do not fully understand the meaning of comparing the empirical results to the PRG on the Ising model (comparison seems to be merely based on the fact that Ising model has a second order phase transition). For example what does the temperature represent for a whole-brain model? Why is the Ising model chosen rather than a different one? In principle one could choose a different model showing a second order phase transition, like the contact process. Then the exponents at the critical point would presumably be different and could for example be more similar to the ones measured in the data for dMRI (instead of EDR)?

► We thank the reviewer for this comment that allows us to justify the choice of the Ising model. Indeed, this choice goes beyond a mere analogy. First of all, using the maximum entropy principle, the Ising model (and its extensions) can be mathematically map to binary data. This has

been done using spiking data (e.g., Schneidman et al. 2006; Tkačik et al. 2014) but also using binarized fMRI data (e.g., Watanabe et al., 2013; Ezaki et al. 2017; Ponce-Alvarez et al., 2022). Briefly, to estimate the probability distribution $P(\boldsymbol{\sigma})$ of binary patterns, i.e., $\boldsymbol{\sigma} = [\sigma_1, \dots, \sigma_N]$, where $\sigma_i = +1$ or -1 if neuron (or ROI) i is active or inactive, respectively, one seeks for the distribution that preserves some statistics of the data and has maximum entropy. For example, it is known that, given some binary data, the maximum entropy distribution that preserves the pairwise correlations of the data is the Boltzmann distribution $P(\boldsymbol{\sigma}) = \exp[-E(\boldsymbol{\sigma})]/Z$, where the pattern's energy is:

$$E(\boldsymbol{\sigma}) = - \sum_{i,j} J_{ij} \sigma_i \sigma_j,$$

$$Z = \sum_{\boldsymbol{\sigma}} \exp[-E(\boldsymbol{\sigma})],$$

where J_{ij} represents the effective interaction between ROIs i and j . We can scale the interactions J_{ij} by a “temperature” parameter β and recover the Boltzmann distribution of an Ising model.

Recently, using maximum entropy models (MEMs), we have shown that the probability distribution estimated from binarized resting-state fMRI data from the macaque is poised close to a critical point, while anesthesia states depart from criticality, towards supercriticality (see: Ponce-Alvarez et al., 2022).

In summary, our previous work (Ponce-Alvarez et al., 2020) suggest that resting-state fMRI dynamics operate close to a phase transition that is given by the Ising model, which is a model that can be mapped to binarized fMRI data through the maximum entropy principle.

We now mention these considerations when we introduce the model (see lines 279-282), in the model's description in Methods (lines 591-600) and in the Discussion (see line 363-364). Finally, see also lines 471-482 for a justification of the use of binarized fMRI data.

8. In Figure 4 the behavior of the model is shown for the critical point as well as at a subcritical and a supercritical point. Several quantities are measured in all 3 regimes and fitted with power laws, however their expected behavior is not power law away from the critical point! Also, in Fig.4D the authors find non-trivial scaling away from the critical point (and a very similar $\tilde{\alpha}$ exponent for the subcritical and the supercritical regimes). This is somewhat surprising and would deserve at least a short discussion.

► We agree that this point was misleading. We now show the goodness of fit of the power laws as a function of the temperature parameter in the new **Supplementary Figure S4**. To evaluate the fit we used the ratio between the explained variance using the power law and the one obtained using an exponential function (R_{EV}).

As shown in **Supplementary Figure S4**, the correlation function, $g(r)$, decays as a power law of distance only around the critical point. On the contrary, the variance of coarse-grained variables, $V(K)$, presents a power-law scaling for all temperatures. This is not surprising because even for uncorrelated or fully correlated data a power law for $V(K)$ is expected, with exponents equal to 1 and 2, respectively (these are trivial cases). Also note that, for the contact process, $V(K)$ present is a power law in different regimes (see Fig 2a in Nicoletti et al., 2020). Finally, the power-law scaling of eigen-spectrum, $\lambda(\text{rank}/K)$, was only observed in the supercritical regime and around the critical point. We now mentioned this in lines 307-310 in the main text.

Accordingly, in **Figs. 5C, 5E, and 5G** of the main text, filled symbols now indicate that R_{EV} favors the power law model, i.e., $R_{EV} > 1$ (note that Fig. 4 in the first submission has become Fig. 5). Also, in **Fig. 5B**, the correlation function (blue) is fitted by a line in the log-log plot only for the critical point.

Minor points:

A. The claim that dMRI connectivity “can be approximated” as an exponentially decaying function is bold and not fully supported by the evidence provided. The authors argue that dMRI data points toward a connectivity matrix where the strength of the interactions decays with (Euclidian) distance between ROIs as a power law, due to long-range (inter-hemispheric) interactions. On the other hand, exponentially decaying connectivities are essentially different network structures, only showing local interactions. I think that the fact that PRG exponents are similar for EDR and dMRI and the fact that a linear model predicts the times series almost equally well for the two structures does not convincingly support the claim that dMRI connectivity “can be approximated” as an exponentially decaying function. I suggest to soften the claim or provide stronger evidence.

▶ As mentioned by the reviewer, our point is that dMRI and EDR predict the time-series almost equally well in a linear model, despite EDR lacking the long-range connections by construction. The connectivity matrices obtained with dMRI and EDR are indeed different. We agree with the reviewer that “can be approximated” was a misleading term. We now removed it; see lines **389-390**.

B. The fit in figure 4F is hard to see due to the fact that the span of the y axis is much larger than the span of the curve.

▶ We now corrected it by changing the range of the y-axis. Please note that Fig. 4 has become Fig. 5.

C. Line 77: the authors claim that the phenomenological renormalization group approach “remains to be tested [...] using theoretical models”. However, as they also point out in the discussion, the method has been applied and its limits of validity have been discussed for the contact process and the Ising models (Nicoletti et al. 2020, cited).

▶ We now cited the work of Nicoletti et al. also in the Introduction. See lines **80-82**.

D. I find the use of the expression “geometry of the brain” in the abstract quite confusing.

▶ We agree that this expression was misleading. We have removed it.

Reviewer #2 (Remarks to the Author):

The paper analyzes resting state fMRI dynamics based on the phenomenological renormalization group method and reports that brain dynamics display power-law correlations. An Ising-type model is used to further distinguish the fMRI results from noise or shuffled data. The paper is clear and well-written, but there seem to be a few points that need to be answered.

1.) The reported power laws are rarely spanning a full decade, although the typical requirement for the acceptance of a power-law interpretation of the data is two orders of magnitude. E.g. scale invariance within less than an order of magnitude is difficult to grasp. I understand that a statistically more solid work would presuppose an unrealistic amount of 1M recordings, which seems an inherent limitation of the empirical renormalization group.

► Indeed, the PRG is limited by finite data. On one hand, the scaling of coarse-grained variables is done as a function of the cluster size K . The range of K is limited by the number of signals in the data. On the other hand, the spectral properties of covariance matrices often depend on the ratio between the number of samples and the number of variables.

For this reason and following the recommendation by Meshulam et al. (2018), we studied cluster sizes for which we have > 10 times more samples than variables. In our case, with 1,200 time frames per scan, we required $K \leq 2^6 = 64$ (a further coarse-graining step would lead to $2^7 = 128$ variables in each cluster, i.e., a number of samples < 10 times larger than the number of variables). We thus coarse-grained the activity up to six times and evaluated the PRG power-laws of $V(K)$ and $F(K)$ for $1 \leq K \leq 64$. The eigen-spectrum was computed for $1 \leq K \leq 64$ and its power-law fit was evaluated for $\text{rank}/K < 0.4$ (similar to Meshulam et al., 2018, 2019) so that the finite-size decay for large relative ranks is not considered. We now mention this in the description of the method (see lines 526-536). We also add some lines on finite data limitation in the Discussion (see lines 380-383).

Nevertheless, notice that power-law fits lead to exponents with low uncertainty. We now report the distribution of the uncertainty (relative estimation error) of $\tilde{\eta}$ and PRG exponents in the new panels Fig. 2C and Fig. 3F. Note that, on average, the uncertainty of exponents was $< 3\%$. See lines 192-196, 204-205, and 233-234.

Finally, to control for sampling effects, we applied the PRG to subsampled data. For this, we built subsampled systems by randomly selecting a fraction of the N ROIs. We found that the PRG exponents from subsampled data converged to those obtained using the full-size original data when the fraction of selected ROIs was larger than $\sim 0.7N$. The results are now shown in Supplementary Fig. S2 and described in lines 242-246.

2.) Both the model (Bialek, Berry and others) and the data analysis (e.g. Hilgetag, Kaiser and others) are not without earlier result of a similar type, which requires some discussion.

► We thank the reviewer for this comment. Indeed, our choice of the spin model is in line with previous work by Bialek and colleagues about maximum entropy models, for example:

- Schneidman E, Berry MJ, Segev R, Bialek W (2006) Weak pairwise correlations imply strongly correlated network states in a neural population. *Nature* 440:1007–1012.
- Tkačik G, Marre O, Amodei D, Schneidman E, Bialek W, Berry MJ II (2014) Searching for collective behavior in a large network of sensory neurons. *PLoS Comput. Biol.* 10:e1003408.

Indeed, using the maximum entropy principle, the spin model (and its extensions) can be mathematically map to binary data. This has been done using spiking data (e.g., Schneidman et al., 2006; Tkačik et al., 2014) but also using binarized fMRI data (e.g., Watanabe et al., 2013; Ezaki et al., 2017; Ponce-Alvarez et al., 2022). Briefly, to estimate the probability distribution

$P(\boldsymbol{\sigma})$ of binary patterns, i.e., $\boldsymbol{\sigma} = [\sigma_1, \dots, \sigma_N]$, where $\sigma_i = +1$ or -1 if neuron (or ROI) i is active or inactive, respectively, one seeks for the distribution that preserves some statistics of the data and has maximum entropy. For example, it is known that, given some binary data, the maximum entropy distribution that preserves the pairwise correlations of the data is the Boltzmann distribution $P(\boldsymbol{\sigma}) = \exp[-E(\boldsymbol{\sigma})]/Z$ (Tkačik et al., 2014), where the pattern’s energy is:

$$E(\boldsymbol{\sigma}) = - \sum_{i,j} J_{ij} \sigma_i \sigma_j,$$

$$Z = \sum_{\boldsymbol{\sigma}} \exp[-E(\boldsymbol{\sigma})].$$

The coupling J_{ij} represents the effective interaction between variables i and j . We can scale the interactions J_{ij} by a “temperature” parameter to β and recover the Boltzmann distribution of our spin model. Recently, using maximum entropy models (MEMs), we have shown that the probability distribution of binarized resting-state fMRI data from the macaque is poised close to a critical point, while anesthesia states depart from criticality, towards supercriticality (Ponce-Alvarez et al., 2022). We now mention these considerations when we introduce the model (see lines 279-282), in the model’s description in Methods (lines 591-600) and in the Discussion (see line 363-364). Finally, see also lines 471-482 for a justification of the use of binarized fMRI data.

Also, we now cite the work of Kaiser and Hilgetag (2004), that uses a model for the probability of a connection between nodes as an exponentially decaying function of the physical distance between them. See lines 89-90.

Kaiser M, Hilgetag CC. Spatial growth of real-world networks. *Phys Rev E Stat Nonlin Soft Matter Phys.* 2004;69:036103.

3.) The paper reports criticality, but does not discuss why this should be expected (or unexpected). 20 years after Beggs and Plenz paper, it seems legitimate to ask about the function and relevant at the particular modality where the data have been recorded.

► We did mention that critical brain dynamics are beneficial for information processing and observed in healthy resting-state data, in the Introduction:

“In physical systems, scale-invariance is observed at critical points. Thus, the observation of power-law statistics in neural activity has contributed to support the idea that spontaneous neural activity operates close to a phase transition (di Santo et al., 2018). Several studies have shown that critical neural systems maximize information transmission, storage, and processing (Shew et al., 2009; Shew and Plenz, 2013; Marinazzo et al., 2014; Ponce-Alvarez et al., 2018, 2022). Interestingly, it has been shown that brain activity deviates from critical-like behavior in different brain states and neuropsychiatric disorders (Hobbs et al., 2010; Meisel et al., 2012; Hesse and Gross, 2014; Tinker and Velazquez, 2014; Massobrio et al., 2015; Cocchi et al., 2017; Rocha et al., 2022). For instance, awake resting state displays critical-like dynamics, while anesthesia states depart from criticality (Tagliazucchi et al., 2016; Fekete et al., 2018; Ponce-Alvarez et al., 2022). For these reasons it is believed that critical dynamics are a characteristic of healthy, awake spontaneous neural activity.”

We have now reformulated this paragraph to highlight those studies that find criticality signatures in resting-state fMRI data. See lines 54-67.

A. Please check the last sentence of the abstract: Validation in a general sense is not possible in a way as described. Also, the "geometry of the brain" is not enough discussed in the paper to be mentioned here.

► We agree that the expression “geometry of the brain” was misleading. We have removed it from the abstract. We also agree that validation is a too strong claim. We thus change “validates” by “tests” in the abstract.

B. Fig 1D: Please find a better way to indicated correspondence between model and data, as the colors are not unambiguous (or there is a problem with my color vision). Fig 1E: The error bar appear to indicate the standard deviation rather than SEM, but in either case the extraction of a nonlinear relation from these data is questionable.

► The reviewer is right: the error bars indicate the standard deviation (SD). We corrected the figure legend. We also changed the color of the trace. The interest of panel Fig. 1E is to show that the linear prediction was similar for nodes with different strengths and was the lowest for those nodes that were weakly connected to the rest of the network (low node strength).

C. 163: Explained variance does not seem to be appropriate to justify a power-law, as the power-law may not even have a variance or the data variance in could be mainly due to the region below some lower cut-off.

► Note that in our study we fitted power laws to scatter data that did not represent probability distributions. The *explained variance* does not refer to the variance of a probability density (which indeed could diverge in the case of a power law probability distribution). Instead, it refers to the coefficient of determination (usually noted R^2 and called “explained variance”) of the regression model we used to fit power laws. We used least squares to fit power laws using the linear regression model $\log Y = a \log X + b$ (i.e., log-log transformation), as in Meshulam et al. (2018, 2019). We evaluated the goodness of fit by comparing the explained variance of the least-squares fit using the power law and the one obtained using an exponential function. Specifically, we calculated the ratio $R_{EV} = R_{PL}^2 / R_{Exp}^2$, where R_{PL}^2 is the explained variance of the linear regression model $\log Y = a \log X + b$ (power law) and R_{EV}^2 is the explained variance of the linear regression model $\log Y = cX + d$ (exponential). Ratios $R_{EV} > 1$ favor the power law hypothesis against the exponential alternative. The estimation error Δa of the power-law exponent was given by the error of the slope a of the linear regression model $\log Y = a \log X + b$. In the new version of the manuscript, we added a description of this fitting procedure in the Methods section; see lines 602-617. We also added the distribution of R_{EV} for the PRG exponents in the new panel 3G.

D. Fig 4B: There is little reason to believe that the curves are straight lines. Instead, it seems to be a subcritical case, where the estimated power-law exponents do not provide additional information. This applies least for the red and the purple curve, whereas the blue curve, which may be the critical case, needs to be represented differently, in its present form no judgement is possible, which is also unacceptable.

► We agree that this point was misleading. We now show the goodness of fit of the power laws as a function of the temperature parameter in the new Supplementary Figure S4. To evaluate the fit we used the ratio between the explained variance using the power law and the one obtained using an exponential function (R_{EV}).

As shown in **Supplementary Figure S4**, the correlation function, $g(r)$, decays as a power law of distance only around the critical point. On the contrary, the variance of coarse-grained variables, $V(K)$, presents a power-law scaling for all temperatures. This is not surprising because even for uncorrelated or fully correlated data a power law for $V(K)$ is expected, with exponents equal to 1 and 2, respectively (these are trivial cases). Also note that, for the contact process, $V(K)$ present is a power law in different regimes (see Fig 2a in Nicoletti et al., 2020). Finally, the power-law scaling of eigen-spectrum, $\lambda(\text{rank}/K)$, was only observed in the supercritical regime and around the critical point. We now mentioned this in lines **307-310** in the main text.

Accordingly, in **Figs. 5C, 5E, and 5G** of the main text, filled symbols now indicate that R_{EV} favors the power law model, i.e., $R_{EV} > 1$ (note that Fig. 4 in the first submission has become Fig. 5). Also, in **Fig. 5B**, the correlation function (blue) is fitted by a line in the log-log plot only for the critical point.

E. Fig 4F: It should be possible to change the scale of the y-axis so that the power-law relation becomes more obvious.

► We agree and we now corrected it by changing the range of the y-axis. Please note that Fig. 4 has become Fig. 5.

FIGURE UPDATES

In the following, we list the changes in main figures that we made in response to the questions of the reviewers. We noted this questions as “Q(reviewer #).(question label)”; for example, Q2.1. denotes the first question of reviewer 2.

Figure 1. The color of the trace in Panel 1E was changed in response to question D of reviewer 2 (noted Q2.D).

Figure 2. Panel C was added in response to Q1.1 and Q2.1. In the first submission, we made errors in the codes generating Panels 2E and 2F, leading to Q1.4. We now have fixed it. We thank the reviewer for noticing the error and we apologize for it.

Figure 3. Error bars in Panels 3A and 3B indicate now SD (instead of SEM, as in the first version), see Q1.1. Panel 3E, 3F and 3G were included in response to Q1.1, Q1.2, Q2.1. and Q2.C.

Figure 4. This is a new figure; see Q1.4.

Figure 5. We added “supercritical” and “subcritical” in Panel 5A for clarity. In Panel 5B, we removed the power-law fits of the red and purple curves, according to Q2.D. In panels 5C, 5E, and 5G, filled symbols now indicate that R_{EV} favors the power law model, i.e., $R_{EV} > 1$, empty symbols indicate $R_{EV} < 1$; see response to Q1.8. and Q2.D. Finally, in Panel 5F, we change the range of the y-axis for better visualization; see Q1.B. and Q2.E.

Reviewers' comments:

Reviewer #2 (Remarks to the Author):

The paper has improved enormously, and it is extremely interesting to consider the relation of dynamical and structural effects in brain dynamics, and given the variety of paradigms in each of these domains, the present paper is a very valuable contribution. However, some of the main questions are still not clear. If the main question here is "EDR or dMRI", then it seems that there is some evidence for their equivalence under certain conditions is given, while a different results appear mainly for the Ising-model approach under the assumption that the brain is poised at criticality (see below).

20: The first sentence of abstract could be reformulated, but needs at least a comma. Also elsewhere in the manuscript the flow and clarity of the text is clearly improvable.

29: "a whole-brain computational models"

30: The claim "we showed that the observed scaling features emerge from critical dynamics and connections exponentially decaying with distance." seem to overstate the actual result that the EDR has higher likelihood than a long-range connectivity derived from dMRI. In line 400, this is stated more clearly. More generally speaking, evidence for EDR could be seen as questionable also conceptually. This ED rule is a strong simplification when compared to the real connectome, and should invite criticism rather than confirmation by comparison to the a set of data that is neither fully reliable. While the evidence against dMRI may hold, the evidence for EDR does not appear sound at this stage.

38: usually → often (?)

77: group → grouped

112: The term "connectivity weight" needs to be explained: Is this an assumption, a biological term, a number resulting from a model? Similarly, it should be stated whether "node strength" is defined in the usual way (lines 133 and 146).

136: Fig. 1B) is nearly impossible to "read". Although it becomes clear that the correlation matrices are non-symmetric, any features that could give an indication of the asymmetry are not discernible. Furthermore, it seems that both dMRI and the EDR have limits when connections across hemispheres are considered, so that it could be interesting to restrict the analysis to single hemispheres (This may also be interesting in regard to potential disparity between the hemispheres.)

207: Fig 3F) What is the meaning of the relative error of an exponent? My feeling is that exponents are more akin to additivity, and larger exponents could be less precise because the higher slope implies the need of a wider range of measurements.

224: Check "were compared the those obtained using".

268: Considering the near perfect correlations in Fig. 4A), a plot of FC-PRG vs. EDR-PRG can be expected to look exactly like Fig. 4B), so I'm don't see the purpose of Fig. 4B in the present context.

325: In Figs. 5 C & E), the exponents do not have a well-defined value near criticality (interestingly the discontinuity occurs at a slightly subcritical case). As we cannot decide whether the brain is poised

exactly to criticality or to a slightly subcritical state, we are cannot know how the discontinuities of these two exponents help to discriminate between the EDR and the long-range connectivity. The μ -exponent, however, appears to favor EDR, but I fail to see that there is actually a power-law spectrum or what could be the significance of the mean-error of 0.001. Merely numerical values for μ are given, but no tests for power-law-ness are included, and from visual inspection of Fig. 3C (or S3C) it does not seem likely that such a test is positive.

Reviewer #2 (Remarks to the Author):

The paper has improved enormously, and it is extremely interesting to consider the relation of dynamical and structural effects in brain dynamics, and given the variety of paradigms in each of these domains, the present paper is a very valuable contribution. However, some of the main questions are still not clear. If the main question here is “EDR or dMRI”, then it seems that there is some evidence for their equivalence under certain conditions is given, while a different results appear mainly for the Ising-model approach under the assumption that the brain is poised at criticality (see below).

20: The first sentence of abstract could be reformulated, but needs at least a comma. Also elsewhere in the manuscript the flow and clarity of the text is clearly improvable.

► We have reformulated this sentence: “Scale invariance is a characteristic of neural activity. How this property emerges from neural interactions remains a fundamental question”.

29: “a whole-brain computational models”

► Corrected. We thank the reviewer for this correction.

30: The claim “we showed that the observed scaling features emerge from critical dynamics and connections exponentially decaying with distance.” seem to overstate the actual result that the EDR has higher likelihood than a long-range connectivity derived from dMRI. In line 400, this is stated more clearly. More generally speaking, evidence for EDR could be seen as questionable also conceptually. This ED rule is a strong simplification when compared to the real connectome, and should invite criticism rather than confirmation by comparison to the a set of data that is neither fully reliable. While the evidence against dMRI may hold, the evidence for EDR does not appear sound at this stage.

► We now toned down the claim by reformulating this sentence as: “[...] we modeled the brain activity using a network of spins interacting through large-scale connectivity and presenting a phase transition between ordered and disordered phases. Within this simple model, we found that the observed scaling features were likely to emerge from critical dynamics and connections exponentially decaying with distance.” See lines 29-32. We believe that the fact that EDR produces many features of the data (linear prediction, correlation function, PRG exponents) is an important result. Also, note that the HCP is one of the largest and standardized imaging datasets (n = 1,003) available nowadays. We now mentioned in the Discussion that future research may study the link between scaling features and connectivity using different data modalities and more realistic models. See lines 408-410.

38: usually → often (?)

► Corrected. We thank the reviewer for this correction.

77: group → grouped

► Corrected. We thank the reviewer for this correction.

112: The term “connectivity weight” needs to be explained: Is this an assumption, a biological term, a number resulting from a model? Similarly, it should be stated whether “node strength” is defined in the usual way (lines 133 and 146).

► We thank the reviewer for this comment. We used standardized methods in Lead-DBS to produce the structural connectomes for the Schaefer parcellation. The connectivity weight $C_{ij} = C_{ji}$ was given by the number of fibers connecting two brain regions. To have values between 0 and 1, we normalized the weights by dividing them by the largest value, i.e., $\max(\mathbf{C})$. See lines 457-471. Also, we now defined the strength of a node; see lines 134-135.

136: Fig. 1B) is nearly impossible to “read”. Although it becomes clear that the correlation matrices are non-symmetric, any features that could give an indication of the asymmetry are not discernible. Furthermore, it seems that both dMRI and the EDR have limits when connections across hemispheres are considered, so that it could be interesting to restrict the analysis to single hemispheres (This may also be interesting in regard to potential disparity between the hemispheres.)

► We acknowledge that panel 1B was misleading: it presented the two matrices (dMRI and EDR) in one single graph. The upper triangular portion corresponded to dMRI, while the lower triangular portion corresponded to EDR. This was confusing and created the impression that matrices were asymmetric, although both dMRI and EDR matrices are symmetric by construction. For clarity, we now presented the matrices separately in the updated figure. We also mentioned that matrices are symmetric; see lines 122-123.

Following the reviewer’s suggestion, in addition to the whole-brain analysis (Fig. 1), we computed:

- i) The linear prediction of the left hemisphere’s activity $\mathbf{X}^{(L)}$ given the corresponding intra-hemispheric connectivity $\mathbf{C}^{(L,L)}$, i.e., $\mathbf{X}_{\text{pred}}^{(L)} = \mathbf{C}^{(L,L)} \mathbf{X}^{(L)}$.
- ii) Same as (i) but for the right hemisphere, i.e., $\mathbf{X}_{\text{pred}}^{(R)} = \mathbf{C}^{(R,R)} \mathbf{X}^{(R)}$.
- iii) The linear prediction of the right hemisphere’s activity $\mathbf{X}^{(L)}$ given the inter-hemispheric connectivity $\mathbf{C}^{(L,R)}$ and the activity of the right hemisphere $\mathbf{X}^{(R)}$, i.e., $\mathbf{X}_{\text{pred}}^{(L)} = \mathbf{C}^{(L,R)} \mathbf{X}^{(R)}$.
- iv) Same as (iii) but for the left hemisphere, i.e., $\mathbf{X}_{\text{pred}}^{(R)} = \mathbf{C}^{(R,L)} \mathbf{X}^{(L)}$.

The connectivity matrix \mathbf{C} was given by the dMRI, the EDR, or a shuffled connectivity that preserves the distribution of dMRI weights but destroys their spatial organization. The goodness of the linear prediction was given by the correlation between $X_{\text{pred}}^{(j)}(t)$ and $X^{(j)}(t)$, where $j = L$ or R , for all nodes and all subjects. The results are presented in the new **Supplementary Figure S1**. We found that, although inter-hemispheric predictions were reduced with respect to intra-hemispheric ones, they remained significant and were practically indistinguishable using the dMRI and the EDR connectivity matrices. See lines 135-141 in the main text and **Supplementary Figure S1**.

207: Fig 3F) What is the meaning of the relative error of an exponent? My feeling is that exponents are more akin to additivity, and larger exponents could be less precise because the higher slope implies the need of a wider range of measurements.

► As mentioned in lines 612-627, the estimation error Δa of the power-law exponent is given by the error of the slope a of the linear regression model $\log Y = a \log X + b$. The relative error of the exponent is given by the relative error of the regression slope: $100 \times \Delta a / a$ (expressed as a percentage). Also, the inverse of the relative error, i.e., $a / \Delta a$, is the t-statistic of the regression slope.

The intuition of the reviewer turns out to be incorrect. Indeed, as shown in the figure below, there's no evidence of correlation between the estimation error and the value of the exponent. Thus, larger exponents are not necessarily less precise.

Figure. Estimation error of the exponent vs. exponent value. Each point represents an individual fMRI scan ($n = 1,003$).

224: Check “were compared the those obtained using”.

► Corrected. We thank the reviewer for this correction.

268: Considering the near perfect correlations in Fig. 4A), a plot of FC-PRG vs. EDR-PRG can be expected to look exactly like Fig. 4B), so I’m don’t see the purpose of Fig. 4B in the present context.

► Note that Fig. 4A compares connectivity-based PRG exponents using the two connectivity matrices, while Fig. 4B compares the FC-based PRG against one connectivity-based PRG (using dMRI). We think that the two panels are complementary. What would be redundant would be to include a third panel comparing the FC-based PRG against the EDR-based PRG.

325: In Figs. 5 C & E), the exponents do not have a well-defined value near criticality (interestingly the discontinuity occurs at a slightly subcritical case). As we cannot decide whether the brain is poised exactly to criticality or to a slightly subcritical state, we are cannot know how the discontinuities of these two exponents help to discriminate between the EDR and the long-range connectivity. The mu-exponent, however, appears to favor EDR, but I fail to see that there is actually a power-law spectrum or what could be the significance of the mean-error of 0.001. Merely numerical values for mu are given, but no tests for power-law-ness are included, and from visual inspection of Fig. 3C (or S3C) it does not seem likely that such a test is positive.

► The reviewer is wrong concerning this point: the exponents are actually well-defined around the critical point. In Figures 5C and 5E filled symbols indicate explained variance ratios favoring the power law model over the exponential model, i.e., $R_{EV} > 1$. Thus, power exponents are well-defined around the critical point for the exponent $\tilde{\eta}$ (Fig. 5C) and for the full range of the temperature parameter for the exponent $\tilde{\alpha}$ (Fig. 5E). The fact that the correlation function $g(r)$ is

a power-law, $g(r) \sim r^{-\tilde{\eta}}$, around the critical point is a known feature of the spin model. Thus, the result that the exponent $\tilde{\eta}$ is only well-defined around the critical point is not surprising. As mentioned in the previous rebuttal, the variance of coarse-grained variables, $V(K)$, presents a power-law scaling, $V \sim K^{\tilde{\alpha}}$, for all temperatures. This is also not very surprising because even for uncorrelated or fully correlated data a power law for $V(K)$ is expected, with exponents equal to 1 and 2, respectively (these are trivial cases). Also note that, for the contact process, $V(K)$ presents a power law in different regimes (see Fig 2a in Nicoletti et al., 2020). Finally, concerning the exponent μ , as for all other exponents, the power-law was tested against an exponential fit for the data and the model; see **Figure 3G** and **Supplementary Figure S4**.

FIGURE UPDATES

Figure 1. Panel 1B was changed: the dMRI and EDR connectivity matrices are presented separately.

Supplementary Figure S1. New supplementary figure.

Reviewers' comments:

Reviewer #2 (Remarks to the Author):

Although the manuscript has dramatically improved, I'm still not sure about the interpretations of the main results (as well as about some of the results).

In particular, Figs. 2A, 3C, and 5B are not unambiguously identifiable as power laws. Thus, I would not consider the phrases "was approximately power-law" (stated for 2A) and "the correlation function is a power law" (stated for Fig. 5B, whereas Fig. 3C is not discussed in the paper which is likewise a deficit) as implied from the figures. Often it is required that two orders of magnitude should be spanned to indicate the presence of a power law, which cannot be considered as generally reliable, but not even this rule holds here. The only way to read the results is thus a statement like: "IF IT WAS A POWER LAW, then the exponent would be in this or this range". Whether this is sufficient to draw a conclusions like "scaling of rs-fMRI activity relates to criticality" depends on the definition of the term "criticality" for which is has been agreed, however, that the presence of power-laws (even if less ambiguous) is not sufficient.

It is known that critical neural behavior can be obtained in neural systems by fine-tuning or by some feedback mechanism largely independent of the network topology, so it seems not unexpected that an exponential connectivity structure (of sufficient range) can support critical dynamics. However, in relation to Fig. 4 it should be discussed how the reliability of the dMRI results depend on the length of the connections, as errors by failing to trace a connection may show an exponential dependency on the length the very close relationship to EDR would be an artifact.

The confidences in Figs. 3 A, B, C (given as +/- 0.001 and +/- 0.002, i.e. < 1% relative error in all cases) do not seem to be in line with the standard deviations that can be read of 3E (around 0.05), and both of them are not in a clear relation with the results in Fig. 3F (μ is more than 4 times smaller than α , its standard deviation μ is about half of that of α , but the relative error is the more or less the same, and in contrast to A, B, C the relative error is now more the 1%). I assume that in A, B, C a standard error is given, which is misleading, while the difference between E and F need more explanation.

439: "We used one rs-fMRI acquisition" implies that the authors obtained the data, but 646 states that the data were taken from a public repository. It is not clear to the reviewer, whether 439 the describes of the repository data, or whether the data were later put in the repository, or whether two different fMRI data sets were used, and which is which.

Reviewers' comments:

Reviewer #2 (Remarks to the Author):

Although the manuscript has dramatically improved, I'm still not sure about the interpretations of the main results (as well as about some of the results).

1) In particular, Figs. 2A, 3C, and 5B are not unambiguously identifiable as power laws. Thus, I would not consider the phrases “was approximately power-law” (stated for 2A) and “the correlation function is a power law” (stated for Fig. 5B, whereas Fig. 3C is not discussed in the paper which is likewise a deficit) as implied from the figures. Often it is required that two orders of magnitude should be spanned to indicate the presence of a power law, which cannot be considered as generally reliable, but not even this rule holds here. The only way to read the results is thus a statement like: “IF IT WAS A POWER LAW, then the exponent would be in this or this range”. Whether this is sufficient to draw a conclusions like “scaling of rs-fMRI activity relates to criticality” depends on the definition of the term “criticality” for which is has been agreed, however, that the presence of power-laws (even if less ambiguous) is not sufficient.

► Note that we tested the power-law hypothesis against an exponential alternative by calculating the ratio between explained variances (R_{EV}) of least-squares fits using the two competing regression models. These ratios systematically favor the power-law model as they were all > 1 for the correlation function and the PRG coarse-grained variables (variance, free energy, and eigen-spectra) for all subjects (Figs. 2D and 3H). Some features of the data limit the range supporting power laws for correlation functions and eigen-spectra.

For the correlation function of the data (Fig. 2A) and the model (Fig. 5B), the range of distances is limited by the brain's volume. However, power-law fMRI correlations have been previously reported (Expert et al., 2011), with a similar exponent (i.e., 0.51, average over seven subjects), and it is also known that power-law correlations are observed for the critical Ising model (blue trace in Fig. 5B). Our results reproduce the observation of Expert et al. (2011) and go beyond by applying the PRG to large-scale fMRI data. We now clarified this in lines 167-168. We also noticed that we made a typo by writing “correlation functionS” in line 311, which could lead to think that all traces in Fig. 5B are power laws, while only the blue one (corresponding to the critical point) is. This explains the concern of the reviewer. We have corrected this typo.

For the eigen-spectrum of the data (Fig. 3C), as described by Meshulam et al. (2018) and in lines 547-553, the number of samples with respect to the number of variables limits the range of ranks which can be studied. However, note that not only the ratio of explained variances is larger than one (although closer to 1 than for the other observables) but also the spectra in clusters of different size collapse when the rank is normalized, i.e., rank/K, consistent with Meshulam et al. (2018). This was mentioned in the methods section (lines 543-546). We now highlighted this when presenting the results in lines 243-248.

Reference:

Expert, P. *et al.* Self-similar correlation function in brain resting-state functional magnetic resonance imaging. *J. R. Soc. Interface*, 8472-479 (2011).

2) It is known that critical neural behavior can be obtained in neural systems by fine-tuning or by some feedback mechanism largely independent of the network topology, so it seems not unexpected that an exponential connectivity structure (of sufficient range) can support critical dynamics. However, in relation to Fig. 4 it should be discussed how the reliability of the dMRI

results depend on the length of the connections, as errors by failing to trace a connection may show an exponential dependency on the length the very close relationship to EDR would be an artifact.

► In Fig. 4 activity variables were coarse-grained using the structural connectivity (for dMRI or EDR). As described in Methods section (lines 562-579), this connectivity-based coarse-graining iterative procedure groups the variables that are maximally connected and accordingly coarse-grains the connectivity matrix. The weights of the updated connectivity matrix at the k -th coarse-grain step result from averaging 4^k connectivity weights from the original connectivity matrix. Because the strongest connections are observed for brain regions which are close in space (see Fig. 1A), the connections between distant brain regions do not play a role in the first coarse-graining step. In further stages of coarse-graining the distances between clusters slowly increase, while the connectivity becomes an increasingly coarse-grained version of the original one (after the 5th coarse-graining step the weights of the updated matrix result from averaging $> 1,000$ original connectivity weights). Thus, hypothetic errors related to the length of connections are averaged out by the coarse-graining procedure.

To clarify this, we added the following remark in the Methods section: “Notice that the weights of matrix $C^{(k)}$ result from averaging 4^k connectivity weights from the original connectivity matrix”; see lines 579-580. Also, we added two panels to **Supplementary Figure S4** showing the distribution of distances between grouped nodes as a function of coarse-graining steps, for both the dMRI and the EDR connectivity matrices.

3) The confidences in Figs. 3 A, B, C (given as +/- 0.001 and +/- 0.002, i.e. $< 1\%$ relative error in all cases) do not seem to be in line with the standard deviations that can be read of 3E (around 0.05), and both of them are not in a clear relation with the results in Fig. 3F (μ is more than 4 times smaller than α , its standard deviation μ is about half of that of α , but the relative error is the more or less the same, and in contrast to A, B, C the relative error is now more the 1%). I assume that in A, B, C a standard error is given, which is misleading, while the difference between E and F need more explanation.

► We now showed the absolute estimation error in the new panel Fig. 3F. The standard deviations of the distributions of exponent values across subjects (Fig. 3E) go in line with the absolute estimation error. We updated the text and the figure legend accordingly; see lines 201-207 and 233.

4) 439: “We used one rs-fMRI acquisition” implies that the authors obtained the data, but 646 states that the data were taken from a public repository. It is not clear to the reviewer, whether 439 the describes of the repository data, or whether the data were later put in the repository, or whether two different fMRI data sets were used, and which is which.

► We indicated that the data comes from the HCP database in section *Materials availability* (see lines 653-659) and in the Reporting Summary document. To make it even clearer, we have rephrased the part in question as:

“In this study we analyzed publicly available rs-fMRI data from the Human Connectome Project (HCP), from 1,003 participants. The participants were scanned on a 3T connectome-Skyra scanner (Siemens). The rs-fMRI data was acquired for approximately 15 minutes, with eyes open and relaxed fixation on a projected bright cross-hair on a dark background. The HCP website (<https://www.humanconnectome.org/>) provides the details of participants, the acquisition protocol and preprocessing of the functional data.” See lines 444-449.

We also edited the text at the beginning of the *Results* section to make it clear: “The data was obtained from the Human Connectome Project (HCP) public database”; see lines 110-111.

FIGURE UPDATES

Figure 3. Panel 3F was added: it represents the absolute estimation error of PRG exponents. Panels 3G and 3H were renamed accordingly.

Supplementary Figure S4. Panels S4F and S4I were added: they show the distributions of distances between grouped nodes as a function of coarse-graining, for the dMRI and the EDR connectivity matrices, respectively. Other panels were renamed accordingly.